# Probiotics reshape the coral microbiome in situ without detectable off-target effects in the surrounding environment
Nathalia Delgadillo-Ordoñez [1,2], Neus Garcias-Bonet [1], Inês Raimundo [1,2], Francisca C. García [1],
Helena Villela[1], Eslam O. Osman[1], Erika P. Santoro[1], Joao Curdia[1], Joao G. D. Rosado[1,2], Pedro Cardoso[1,2],
Ahmed Alsaggaf[1,2], Adam Barno[1,2], Chakkiath Paul Antony[1], Carolina Bocanegra[1],
Michael L. Berumen [1,2], Christian R. Voolstra [3], Francesca Benzoni[1,2], Susana Carvalho[1,2] &
Raquel S. Peixoto [1,2] ✉

Beneficial microorganisms for corals (BMCs), or probiotics, can enhance coral resilience against
stressors in laboratory trials. However, the ability of probiotics to restructure the coral microbiome in
situ is yet to be determined. As a first step to elucidate this, we inoculated putative probiotic bacteria
(pBMCs) on healthy colonies of *Pocillopora verrucosa* in situ in the Red Sea, three times per week,
during 3 months. pBMCs significantly influenced the coral microbiome, while bacteria of the
surrounding seawater and sediment remained unchanged. The inoculated genera *Halomonas*,
*Pseudoalteromonas*, and *Bacillus* were significantly enriched in probiotic-treated corals. Furthermore,
the probiotic treatment also correlated with an increase in other beneficial groups (e.g., *Ruegeria* and
*Limosilactobacillus*), and a decrease in potential coral pathogens, such as *Vibrio*. As all corals (treated
and non-treated) remained healthy throughout the experiment, we could not track health
improvements or protection against stress. Our data indicate that healthy, and therefore stable, coral
microbiomes can be restructured in situ, although repeated and continuous inoculations may be
required in these cases. Further, our study provides supporting evidence that, at the studied scale,
pBMCs have no detectable off-target effects on the surrounding microbiomes of seawater and
sediment near inoculated corals.

Coral reefs are important ecosystems in the marine environment, supporting a wide range of organisms and providing essential ecosystem services to society[1–3]. Despite their ecological and economic importance, it is estimated that by 2030 approximately 60% of coral reefs worldwide will be under threat due to global (e.g., ocean warming due to increased $CO_2$ emissions) and local impacts (e.g., water pollution, coastal development, and overfishing)[4,5]. While mitigating ocean warming and local impacts is crucial for preserving coral reefs[6,7], there has been a growing focus on active intervention strategies aimed at enhancing the natural resilience of corals to cope with the already established negative effects of environmental stressors[2,8,9]. One strategy is to use beneficial microorganisms for corals (BMCs, or coral probiotics) to rehabilitate the coral microbiome, promoting

coral health[10,11]. In aquarium experiments, probiotics have shown beneficial effects in promoting coral growth and mitigating the effects of pollution, bleaching, and disease, even preventing coral mortality[12–18]. This approach involves isolating coral-associated bacteria with subsequent screening for beneficial traits for the coral holobiont and developing customized probiotic cocktails for corals[10,19].

The "coral probiotic hypothesis" suggested that microbial communities in the coral mucus and tissue layers act as a defense against pathogens and assist in rapid acclimation to changing environmental conditions[20,21]. Later studies provided further evidence of the beneficial interactions between a diverse community of coral-associated bacteria and the coral holobiont, through mechanisms such as nutrient cycling, production of

[1]Red Sea Research Center, Biological and Environmental Science and Engineering Division, King Abdullah University of Science and Technology, Thuwal, Saudi Arabia. [2]Marine Science and Bioscience Programs, Biological and Environmental Science and Engineering Division, King Abdullah University of Science and Technology, Thuwal, Saudi Arabia. [3]Department of Biology, University of Konstanz, Konstanz, Germany. ✉e-mail: raquel.peixoto@kaust.edu.sa

antimicrobial substances, degradation of sulfur substances and toxic compounds, scavenging reactive oxygen species (ROS), and facilitation of coral larval settlement (e.g., refs. [22–34]). These and additional putative mechanisms were included within a framework proposing the active use of these beneficial traits as probiotics [10,11].

Since the proof of concept shows that BMCs can improve coral health and mitigate damage caused by oil spills, pathogen infection, and thermal bleaching [12,13], further research has explored the underlying mechanisms, physiological effects, and application of BMCs across different coral species and life stages (e.g., refs. [14–18,35–38]). Recent advancements include an enhanced understanding of BMC mechanisms, the selection of probiotic strains based on genomic screening, and the exploration of other microbial therapies and bacterial experimental evolution [19,39–42].

Despite these promising developments, the ability to restructure the coral microbiome using probiotics in situ was not previously tested, and the overall effect of BMC administration on the coral-associated microbiome and surrounding reef ecosystem remains unclear [19]. In this study, we conducted repeated in situ probiotic inoculations three times per week, over a 3-month period on colonies of the scleractinian *Pocillopora verrucosa* to investigate changes in their microbiome, surrounding microbial communities and coral health. Three of the inoculated pBMC genera were enriched in the coral microbiome, which was aligned with an overall restructuring of the coral microbiome. Importantly, the probiotic application did not affect the bacterial communities in the surrounding seawater and sediment. This study provides the first experimental evidence of the feasibility of restructuring healthy coral microbiomes using probiotics in the ocean without affecting the microbial communities of the surrounding environment.

## Results

### Probiotic consortium selection and assemblage

The putative probiotic consortium (here referred to as pBMCs strains) was composed of six bacterial strains isolated from visually healthy colonies of *P. verrucosa* (two *Pseudoalteromonas galatheae* and two *Cobetia amphilecti*), *Stylophora pistillata* (one *Halomonas* sp.), and *Galaxea fascicularis* (one *Suctlifiella* sp.) collected in the central Red Sea. Putative BMC strains were selected based on exhibiting at least one of the following assumed beneficial traits via in vitro testing: antagonistic effect against the coral pathogen *Vibrio corallilyticus* (measured through the diffusion agar method) [43,44], ROS scavenging (measured through catalase activity), which potentially minimizes ROS concentration during thermal stress [10]; production of siderophores (measured through siderophores excretion), which bind to iron compounds and increase their concentration, making them into bioavailable forms for supporting *Symbiodinaceae* metabolism [45,46]; phosphate assimilation (measured through positive activity of phosphate-solubilizing bacteria) to support the coral metabolism [2,47]; and urease activity, (measured through urease secretion to hydrolyze urea), to support nitrogen cycling by making bioavailable nitrogen compounds for the coral holobiont [48,49]. From the six selected pBMCs strains, the two *P. galatheae* (30H and 31H) were positive for catalase, *C. amphilecti* strains (65H and 81H) were positive for catalase and phosphate assimilation, *Halomonas* sp. (SAT10) was positive for catalase and siderophores production, and *Sutcliffiella* sp. was positive for catalase and siderophores production. (Supplementary Table 1). We conducted a 3-month probiotic inoculation (three times per week) in situ on visually healthy colonies of *P. verrucosa* in the central Red Sea (T1–T3). We monitored the bacterial community of the coral before (T1), during (T2), and at the end of the probiotic inoculations (T3), as well as 5 months after the last inoculation (T4), covering seasonal variations (represented here as seasonal seawater temperature changes in the study site). We also assessed changes in nearby water and sediment bacterial communities before and after the last inoculations (T1 and T3) through 16S ribosomal RNA (rRNA) gene amplicon sequencing. Additionally, we monitored the corals' physiology by measuring the photosynthetic efficiency of the *Symbiodiniaceae* present within the coral tissue in situ (T1–T4) and the corals' heat response in CBASS-controlled experiments (T1–T4) (Fig. 1 a–d) (see details in "Methods").

### Effect of the probiotic in situ inoculation on the coral microbiome and surrounding environment

We assessed the effect of coral probiotics (pBMCs) on the microbiome of visually healthy *P. verrucosa* colonies by monitoring the bacterial community at four sampling points (T1, T2, T3, and T4) interspersed between inoculations. Amplicon 16S rRNA gene sequencing revealed that the long-term probiotic inoculation in situ significantly changed the coral microbiome of the microbiome regulator (i.e., coral species that usually maintain a constant microbiome) *P. verrucosa* [50] ($Adonis_{(treatment)}$, $R2 = 0.018$, d$f = 1$, $F = 2.3498$, $Pr (>F)$ 0.021). Specifically, the bacterial community of probiotic-treated corals only differed from placebo-treated corals after the last inoculation period (T3) ($Adonis_{(treatment)}$, $R2 = 0.137$, d$f = 1$, $F = 4.2909$, $Pr (>F)$ 0.002) and not at the other sampling times (Fig. 2 a–d). Similarly, in T3, probiotic-treated corals showed a significant increase in alpha-diversity metrics compared to placebo-treated corals: (Shannon (H'), Simpson, and Chao1 ($Wilcox$, $p$ values: < 0.00028; 0.0079; < 0.0001, respectively) (Fig. 3a). We also detected an enrichment in ASVs belonging to the same genera of the inoculated pBMCs (Fig. 3b), where comparisons between probiotic-treated and placebo samples revealed significant increases in the relative abundance of *Halomonas* and *Pseudoalteromonas* in probiotic-treated corals compared to placebo-treated corals ($Wilcox$, $p$ values 0.04 and 0.001, respectively). However, *Cobetia* was not detected in T3 in either treatment, while the genus *Sutcliffiella* was not detected at any sampling time. Nonetheless, *Bacillus*, the former taxonomic classification of *Sutcliffiella*, was also used as a proxy for this genus (see "Methods"), and was significantly enriched in probiotic-treated corals ($Wilcox$, $p$ value 0.03) (Fig. 3b). In addition, the coral microbiome changed over time ($Adonis_{(sampling\ time)}$, $R2 = 0.106$, d$f = 3$, $F = 4.5125$, $Pr (>F)$ 0.001) (Supplementary Fig. 1), and the interaction between treatment and sampling time was not statistically significant ($Adonis_{(treatment \times sampling\ time)}$, $R2 = 0.038$, d$f = 3$, $F = 1.6347$, $Pr(>F)$ 0.055).

To further explore the compositional differences, the main bacterial community groups at sampling time T3 were investigated. We observed differences in the relative abundance of several groups of the dominant taxa between probiotic-treated and placebo-treated corals at T3. Overall, four of the ten most abundant families were significantly enriched in the probiotic-treated corals: *Rikenellaceae*, *Prevotellaceae*, *Lachnospiraceae*, and *Rhodobacteraceae* (Wilcox, $p$ values < 0.0001, < 0.0001; < 0.0001 and 0.00042, respectively). On the contrary, the relative abundance of *Endozoicomonadaceae* decreased in probiotic-treated corals ($Wilcox$, $p$ value = 0.0068) (Fig. 4a); other families like *Simkaniaceae*, *Burkholderiaceae*, Unclassified Alphaproteobacteria, *Spirochaetaceae*, and *Phycisphaeraceae* did not significantly differ between treatments (Supplementary Table 2). Results revealed a high number of differentially abundant ASVs between probiotic-treated compared to placebo-treated colonies affiliated to several taxonomic groups (ANCOM-BC2, $n = 1175$ ASVs, $p$-$adj$. < 0.05) that were enriched ($n = 245$) or decreased ($n = 930$) in probiotic-treated colonies (Fig. 4b). The most enriched ($p$-$adj$. values < 0.01, $W$ statistic > 4) and decreased ASVs ($p$-$adj$. values < 0.01, $W$ statistic < − 4) mainly belonged to the phylum Proteobacteria ($n = 9$, respectively), and included genera such as *Endozoicomonas* ($n = 4$, respectively), *Salinarimonas* ($n = 1$, enriched), *Delftia* ($n = 1$, enriched), MND1 ($n = 1$, enriched; $n = 2$, decreased), and *Catenococcus* ($n = 1$, decreased). Other groups from the phyla Firmicutes, Nitrospira, Bacteroidota, and Verrucomicrobiota also included differentially abundant ASVs (Fig. 4c). Interestingly, ASVs enriched in probiotic-treated corals showed consistent relative abundances across biological replicates (Fig. 5). Additionally, other enriched ASVs (ANCOM-BC2, $p$-$adj$. < 0.05, $W$ statistic ranging from 2 to 4) included potentially beneficial groups such as members of *Rhodobacteraceae* (i.e., *Ruegeria*), *Lactobacillaceae* (i.e., *Limosilactobacillus*), *Desulfovibrionaceae* (i.e., *Desulfovibrio*), *Lachnospiraceae* (i.e., *Butyrivibrio*), *Oscillospiraceae* (i.e., *Ruminococccus*), and *Nocardiaceae* (i.e., *Rhodococcus*). Unclassified ASVs are detailed in

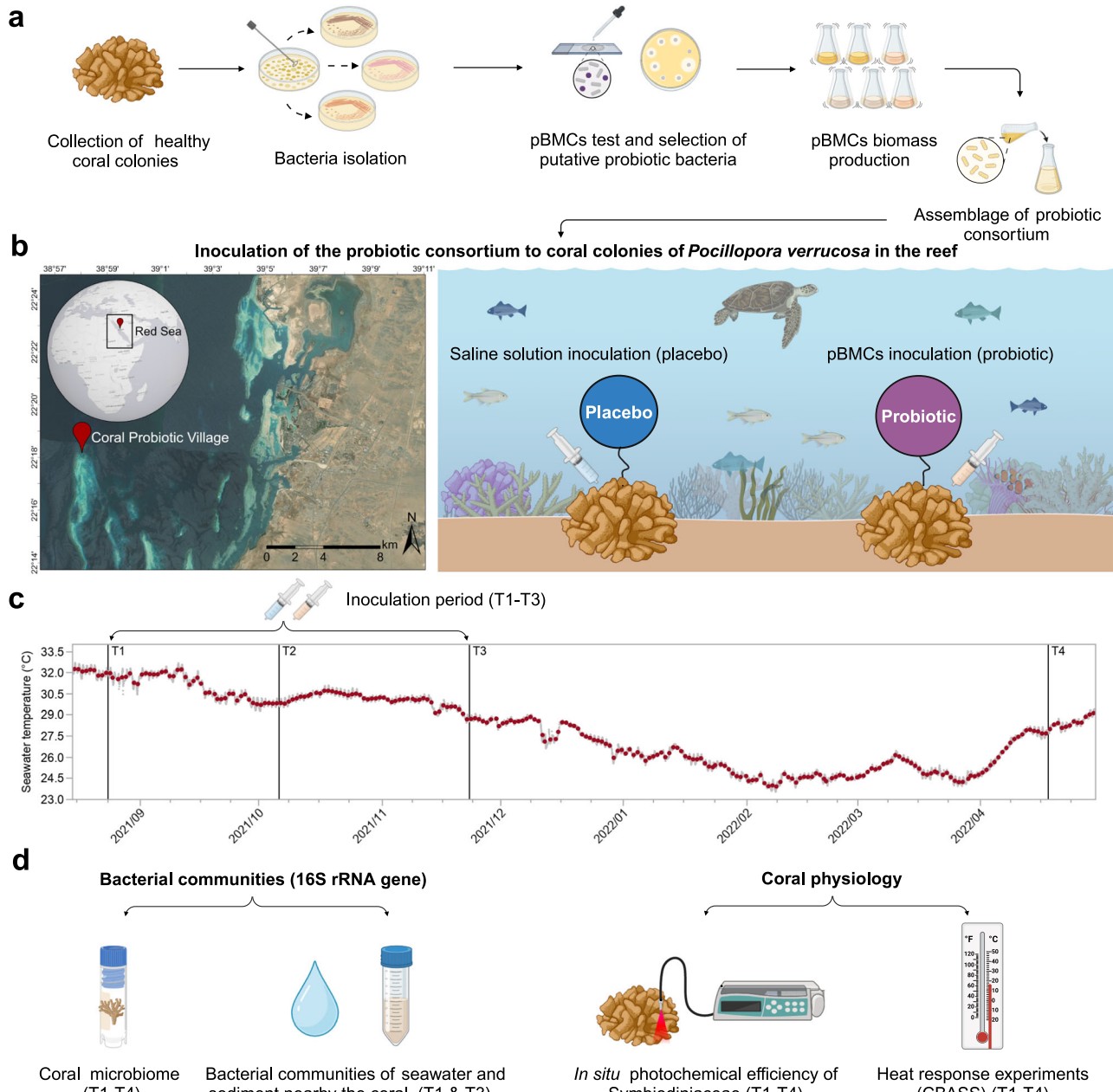

**Fig. 1 | Schematic representation of the experimental design. a** Summary of pBMCs isolation and probiotic assemblage. **b** Location of the study site in the Central Red Sea where in situ experimental treatments took place. The map was created with a licensed version of ArcGIS Pro. Version 2.8.0. **c** Seawater temperature changes during the experiment sampling times. Daily mean temperatures (red dots) are indicated. The inoculation period from T1 to T3 (placebo and probiotic syringes) and all sampling times (T1–T4) are indicated. **d** Summary of analyzed microbial communities (coral, seawater, and sediment near the corals) and monitoring of the coral physiology. Icons indicate the analysis conducted at each sampling time. Infographics were created with a licensed version in BioRender.com.

Supplementary Table 3. Additionally, we evaluated changes in significant ASVs (ANCOM-BC2, *p.adj.* < 0.01) affiliated with the family *Vibrionaceae*, as it includes opportunistic coral pathogenic bacteria. The results revealed a significant decrease in ASVs among probiotic-treated colonies, including *Photobacterium*, *Vibrio*, Unclassified *Vibrionaceae*, and *Catenococcus* (Fig. 4d).

In contrast to the changes observed in corals, the bacterial communities associated with the surrounding seawater and sediment were not affected by the probiotic inoculation when comparing samples collected near coral colonies under different treatments before (T1) and after (T3) the probiotic inoculation period (see details in "Methods"). As the pBMCs can naturally occur in the reef environment, we focused on the overall effect of the

treatment in the bacterial community structure of seawater and sediment. More specifically, the seawater bacterial community did not differ between seawater surrounding probiotic-treated and placebo-treated coral colonies but changed over time ($Adonis_{(treatment)}$ $R2 = 0.04$, d$f = 1$, $F = 2.1104$, $Pr$ (>$F$) 0.105; $Adonis_{(sampling\ time)}$ $R2 = 0.6$, d$f = 1$, $F = 31.444$, $Pr$ (>$F$) 0.001) (Fig. 6a). Similarly, the bacterial community in the sediments was not significantly changed by treatment, yet remained stable over time ($Adonis_{(treatment)}$ $R2 = 0.07$, d$f = 1$, $F = 1.4124$, $Pr$ (>$F$) 0.055; $Adonis_{(sampling\ time)}$ $R2 = 0.06$, d$f = 1$, $F = 1.1946$, $Pr$ (>$F$) 0.181) (Fig. 6b). According to non-metric multidimensional scalingn (MDS) ordination, bacterial communities of coral, seawater, and sediment were distinct, forming three different groups (Supplementary Fig. 2).

**Fig. 2 | Compositional changes in the bacterial community of *Pocillopora verrucosa* associated with the in situ inoculation of coral probiotics.** Nonmetric multidimensional scaling ordination (nMDS) of the *P. verrucosa* microbial community according to sampling time and treatment (k = 2) in **a** T1, **b** T2, **c** T3, and **d** T4. Each point represents a biological replicate (*n*). Biological replicates per sampling time and treatment (*n* = 15).

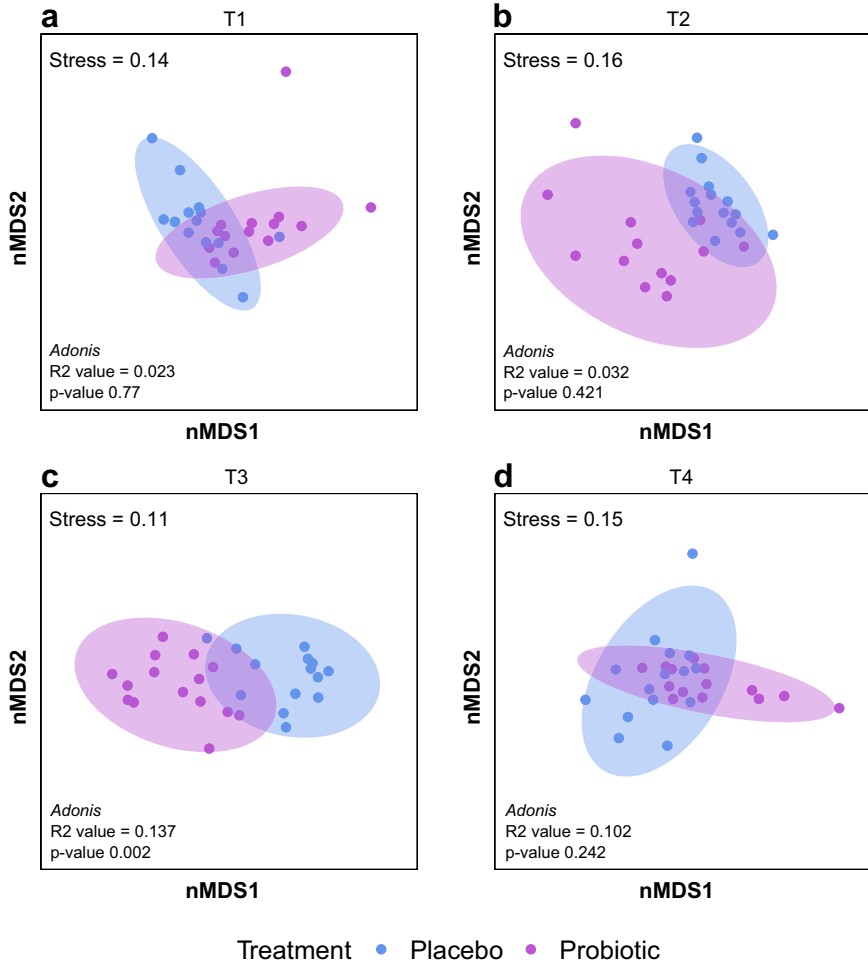

## Coral physiology

The photosynthetic efficiency of *Symbiodiniaceae* and the thermal tolerance were also monitored, although no bleaching event or disease affected the studied corals during the experiment. Thus, the putative benefits of the probiotic treatment could not fully be exerted. Photosynthetic efficiency was measured in situ using a diving PAM ($F_v/F_m$). We did not observe any treatment effect on the coral performance (Fig. 7a and Supplementary Table 4). Although all corals were healthy at all sampling points, significant changes in the $F_v/F_m$ rates were observed over time (*p* value < 0.001) (Supplementary Table 5). The only case where we did not detect differences was between T1 and T4 (*p* value = 0.38).

Standardized thermal tolerance thresholds (ED50s) of *P. verrucosa* were evaluated experimentally at each sampling time using the Coral Bleaching Automated Stress System (CBASS)[51] (see details in the "Method" section). Similar to the coral fitness results, the *P. verrucosa* thermal threshold changed over time points with maximum ED50 values measured in T1 (38.06 ± 0.65) and minimum values in T4 (35.48 ± 0.7) (*p* value < 0.001), with significant changes between time points observed (except between sampling times T2 and T3 [*p* value = 0.71, Supplementary Table 6]), but not between treatments at any given time point (Fig. 7b and Supplementary Table 7).

## Discussion

Here we demonstrated that pBMCs can instigate a restructuring of the stable microbiome of healthy corals in situ without causing permanent changes in the microbial communities of the surrounding environment. Continuous probiotic application led to shifts in the taxonomic composition and diversity of the healthy (and therefore robust[52]) microbiome of *P. verrucosa*, which has also been previously shown to be very resistant to changes in their

microbiome[50]. Still, the continued inoculations promoted an enrichment of the bacteria corresponding to the same genera of some of the inoculated pBMCs. This may suggest that probiotics can be incorporated by corals in situ and/or trigger bacterial enrichment in the coral microbiome. Although the inoculated genera were enriched and we identified the ASVs in the coral microbiome matching 100% similarity with the inoculated pBMCs, those ASVs were eventually found absent or in extremely low abundance in the 16S amplicon data. This does not indicate "lack of success": even in the absence of long-term manifestation of the probiotic strains at high abundance, microbial transfer can still trigger a "reboot" or "reset" of the microbiome that leads to a potentially more beneficial microbiome (as reviewed in ref.[53]). Thus, the restructuring of the microbiome following probiotic treatment, in addition to (ideally) finding the treated bacterial strains in the restructured microbiome, are both indicators of successful treatment.

The bacterial community of *P. verrucosa* was dominated by the family *Endozoicomonadaceae* across seasons and treatments. However, the probiotic treatment led to a decrease in the relative abundance of *Endozoicomonadaceae*, possibly due to the increase of other bacterial groups. *Endozoicomonadaceae* are typically associated with healthy corals[54-63], and are often dominant in the *P. verrucosa* microbiome[50,57]. Their abundance tends to decrease during thermal stress and bleaching[64]. Nonetheless, recent studies suggest that different dominant species of *Endozoicomonadaceae* may have distinct roles and respond differently to local environmental fluctuations[60,65], and can also be associated with bleached corals[57]. As the beneficial role of *Endozoicomonas* is yet to be proved, and we did not observe any negative effect on the coral health upon the probiotic inoculations (discussed later), we argue that the decrease in *Endozoicomonadaceae* did not reflect in a detrimental effect on the coral holobiont, and may have led to

**Fig. 3 | Alpha diversity and relative abundance of pBMCs inoculated genera in the microbiome associated to probiotic-treated corals vs placebo-treated corals. a** Alpha diversity indices (H' = Shannon–Weaver diversity, Simpson, and Chao1) estimated by treatment (placebo and probiotic) at each sampling time are shown. The statistically significant differences are denoted with asterisks: \*\*$p < 0.01$; \*\*\*$p < 0.001$. **b** Relative abundance of the pBMCs genera in the coral microbiome of *P. verrucosa* at T3 according to treatment (placebo and probiotic). Significant differences were detected between treatments in *Bacillus* (here as a proxy of *Sutclifiella*), *Halomonas*, and *Pseudoalteromonas* in probiotic-treated corals. The statistically significant differences are denoted with asterisks: \*$p < 0.05$; \*\*$p < 0.01$; and \*\*\*$p < 0.001$. The depicted boxplots show the median (center line) and the first and third quartiles (lower and upper bounds). Biological replicates per treatment ($n = 15$).

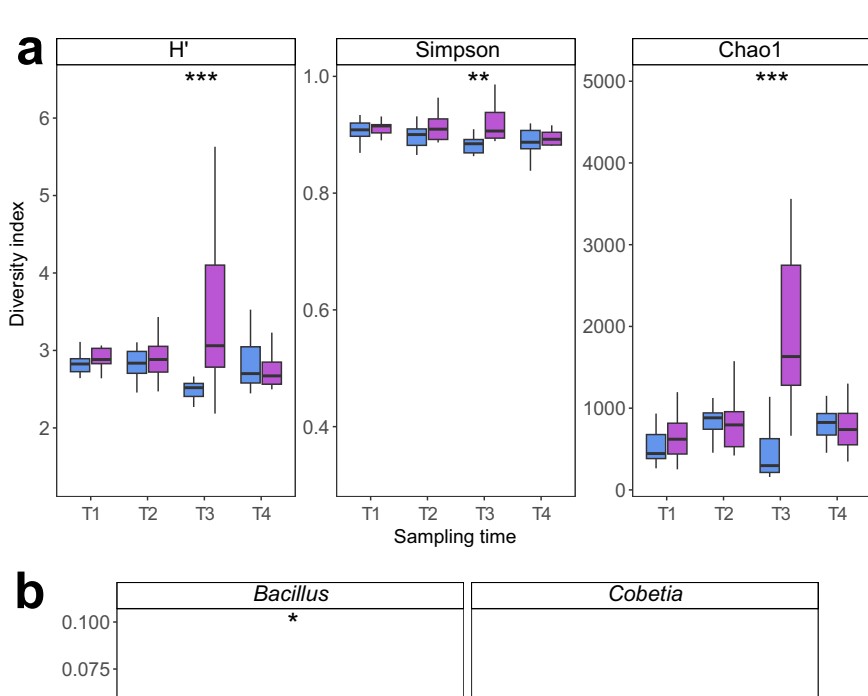

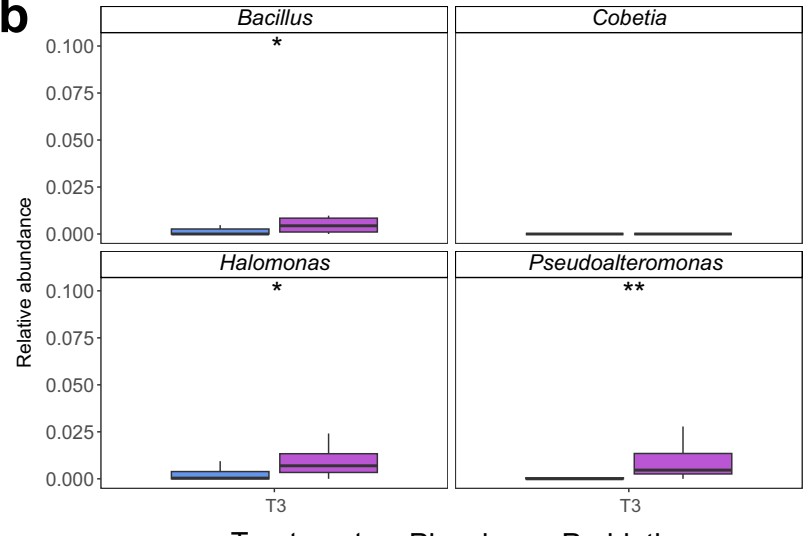

an enrichment of other key bacterial groups (detailed hereafter) that could increase the holobiont resilience in the event of environmental impacts.

For instance, in addition to the enrichment of ASVs belonging to the same genera of the inoculated pBMCs that were previously validated as beneficial for corals (i.e., *Halomonas*, *Bacillus*, and *Pseudoalteromonas*)[13,14], families including *Rikenellaceae*, *Prevotellaceae*, *Lachnospiraceae*, and *Rhodobacteraceae* were also enriched in probiotic-treated corals. Some of these families have been consistently associated with healthy hard (e.g., *Rikenellaceae* and *Prevotellaceae*)[66], and soft corals (order Alcyonaceae) (e.g., *Lachnospiraceae*)[67]. *Rhodobacteraceae* is a common member of coral microbiomes correlated with various health statuses[61,66,68]. Although their role in the coral holobiont is unclear, they seem to be involved in key functions that can promote coral health, including nitrogen cycling, toxic compound degradation, antimicrobial activity[69,70], and also being commonly found associated with the mucus of coral larvae[35]. Some ASVs significantly enriched in probiotic-treated corals are potentially beneficial for the coral holobiont. Some examples include *Simkania* (*Simkaniaceae*), a coral endosymbiont occurring in close association with *Endozoicomonas* bacteria[71]; *Delftia* (*Commamonadaceae*), a key member of coral microbiomes[72] that plays roles in anti-quorum sensing and antibiofilm activity[73] and may help to control pathogenic microbes associated with bleaching[74]; and *Ruegeria* (*Rhodobacteraceae*), known for their role in antimicrobial effects against coral pathogens[75], colonization of early life

stages of coral[76], and degradation of toxic compounds[70]. Other examples include fermentative bacteria such as *Limosilactobacillus* (*Lactobacillaceae*), formerly classified as *Bacillus*, isolated from healthy coral mucus[77], capable of forming stable associations with probiotic bacteria from the genus *Lactobacillus*[78]. Other fermentative bacteria include *Rikenellaceae* RC9 gut group, *Saccharofermentas*, *Ruminococcus*, *Pseudobutyrivibrio*, *Butyrivibrio*, and *Christensenellaceae* R-7 group, which may play a key role in carbon metabolism and nitrogen cycling within the coral holobiont[63,79,80]. They potentially contribute to the degradation of complex carbohydrates (i.e., starch) produced by *Symbiodiniaceae*[63,81]. Some nitrogen-fixing bacteria were also enriched, including *Rhodococcus* (*Nocardiaceae*), also known for its antimicrobial activity[82,83] and degradation of emergent contaminants in the marine environment[84,85]. Other nitrifiers included Mle-1-7 group (*Nitrosomonadaceae*) and *Nitrospira* (*Nitrospiraceae*), which may be important for the primary productivity of coral photosynthetic symbionts by making nitrogen compounds available[86]. We also observed the enrichment of coral intracellular protozoan endosymbionts, such as *Candidatus* Amoebophilus[87–89], which interact with eukaryotic hosts, such as *Symbiodinium* spp. and apicomplexans[90,91].

Moreover, we observed a decrease in bacteria from the family *Vibrionaceae*, which encompasses a wide range of marine bacteria, extensively associated with coral microbiomes[66]. Some *Vibrionaceae* members constitute opportunistic coral pathogens, likely contributing to coral

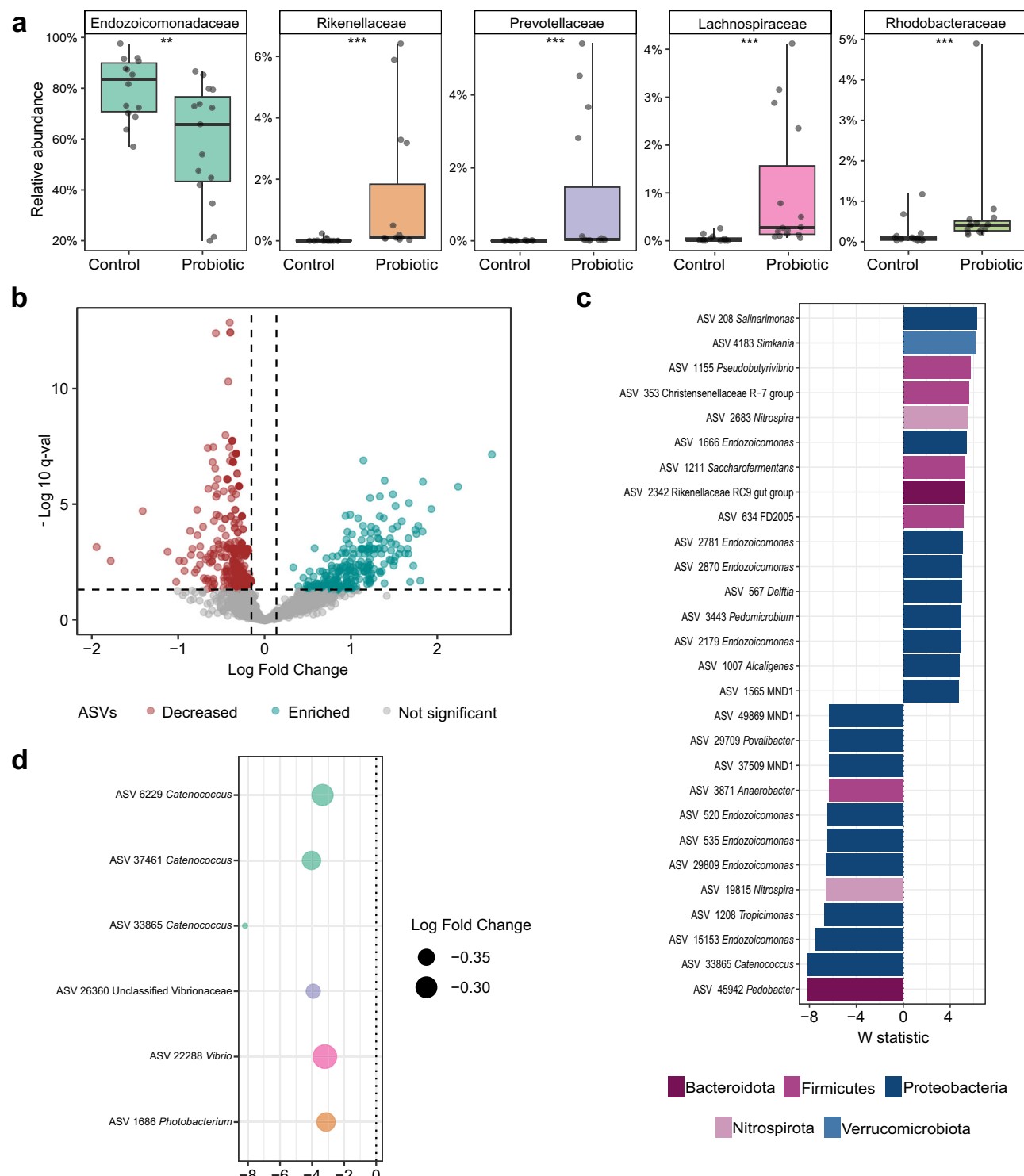

**Fig. 4 | *Pocillopora verrucosa* microbiome restructuring after long-term probiotic inoculation (T3 sampling time). a** Families among the top ten most abundant in the coral microbiome, with significant changes in relative abundance between placebo- and probiotic-treated corals. The statistically significant differences are denoted with asterisks: **p < 0.01; ***p < 0.001. The depicted boxplots show the median (center line) and the first and third quartiles (lower and upper bounds). **b** Volcano plot depicting differentially abundant ASVs (dots) identified in the ANCOM-BC2 analysis (n = 1175 differentially abundant ASVs in probiotic- in comparison to the placebo-treated). The log fold change (X-axis) and the p-adj. (Y-axis) value for each ASV is represented. Blue dots indicate enriched ASVs and red dots indicate decreased ASVs with a q value (adjusted p value) < 0.05. ASVs that are not significantly different in abundance between treatments (probiotic vs placebo) are colored in gray. **c** Top differentially abundant ASVs with their associated genus taxa and color-coded by phylum. The W test statistic from the ANCOM-BC2 is shown (negative W indicates decreased taxa while positive W indicates enriched taxa in probiotic-treated corals in comparison to placebo-treated corals. "Unclassified" ASVs are available in Supplementary Table 3. **d** Differentially abundant ASV affiliated with the *Vibrionaceae* family, colored by genera. The dot size represents the Log Fold change of each ASV. Biological replicates per treatment (n = 15).

**Fig. 5 | Heatmap representing the relative percentage of the top 20 most enriched (top part) and 20 most decreased (bottom part) ASVs in probiotic- vs placebo-treated corals in T3.** The *X*-axis represents biological replicates (colony) for each treatment, respectively. Biological replicates per treatment (*n* = 15).

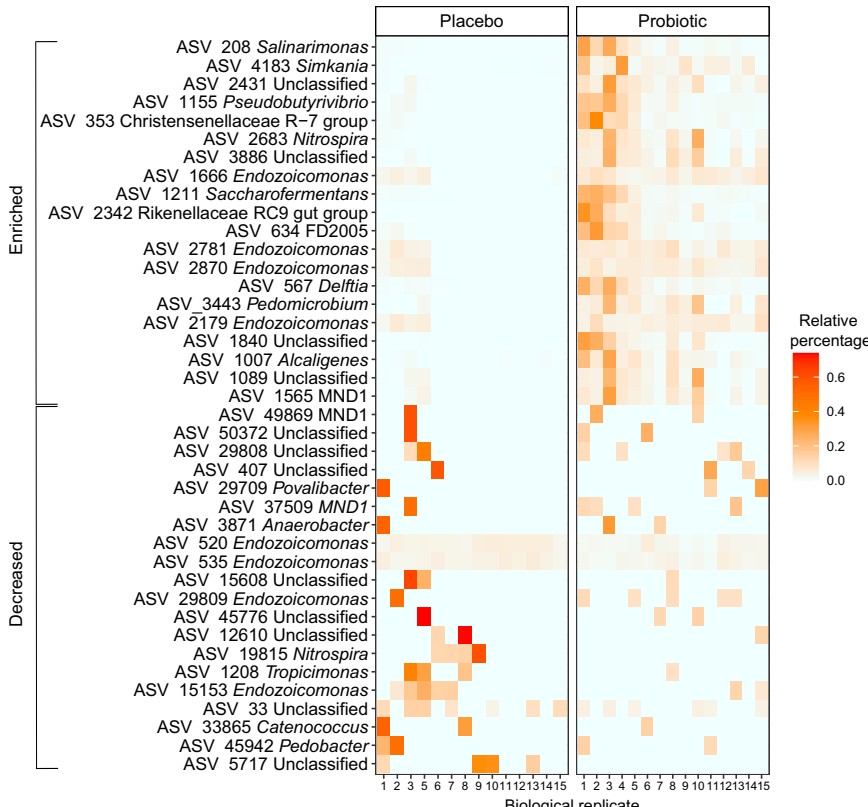

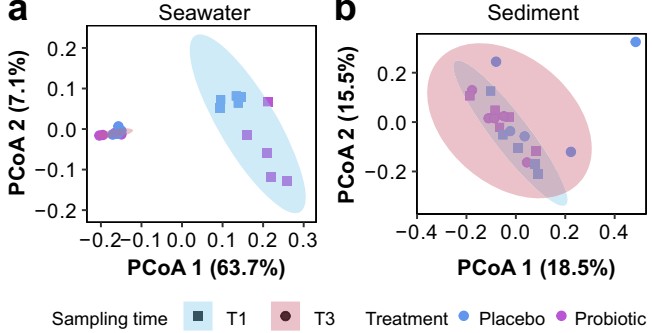

**Fig. 6 | Coral surrounding bacterial communities according to treatments and time.** Principal Component Analysis (PCoA) of **a** seawater and **b** sediment bacterial communities in the surrounding environment of placebo-treated (blue) and probiotic-treated (purple) corals in T1 and T3. Biological replicates per sampling time and treatment (*n* = 5).

diseases and bleaching[43,44,92–99]. We observed that the probiotic treatment led to a decrease in the abundance of *Vibrio* spp., Unclassified *Vibrionaceae*, *Photobacterium*, and *Catenococcus*. Previous studies have shown that probiotics can reduce *Vibrio* abundance and mitigate coral bleaching[13,14]. Nonetheless, it is important to note that some Vibrios can be non-pathogenic[69], and therefore, further studies exploring the restructuring of *Vibrionaceae* with targeted approaches would help enlighten specific shifts in pathogenic and non-pathogenic coral-associated bacteria.

We also explored the aforementioned presence and relative abundance of the ASVs corresponding to the same genera of the inoculated pBMCs in the coral microbiome. We observed an overall enrichment of *Halomonas* and *Pseudoalteromonas* in probiotic-treated corals in T3, which may lead to an increase in beneficial microbial functions in the coral holobiont[13,14]. Although we did not detect the presence of the pBMC *Sutcliffiella*, the current 16S amplicon data pose difficulties in the taxonomic resolution of recently re-classified groups, such as *Sutcliffiella*, a novel genus previously

classified as *Bacillus* and recently described[100]. Nonetheless, we observed enrichment of *Bacillus* in probiotic-treated corals in T3, although there is no certainty of which specific *Bacillus* ASVs correspond to *Sutcliffiella* in the coral microbiome. Previous studies found evidence of successful BMCs incorporation and further enrichment in the coral microbiome upon microbiome manipulation in controlled experiments[13,14,35,38]. In this context, despite the low microbiome flexibility described for the *Pocillopora* genus[50], prokaryotes can also be acquired from the environment[59,76,101], which may have allowed the pBMCs incorporation and/or enrichment. pBMCs genera were found in very low abundance in the native microbiome of the studied *P. verrucosa*, but their relative abundance was significantly increased in T3, suggesting their incorporation and/or enrichment upon frequent probiotic inoculations, even in corals that were not under stress. On the contrary, the enrichment of probiotics seems to be facilitated when corals are under stress[14], which may indicate that, if the goal is the rehabilitation and retention of threatened corals, the use of probiotics as medicine applied in times of stress may be a good strategy to be tested[10,102]. Furthermore, the dominance of certain bacteria, (i.e., *Endozoicomonadaceae*) and the low flexibility of the host microbiome[57] may also influence this process. The use of dominant symbiotic bacteria (i.e., *Endozoicomonas* sp.) may represent a promising strategy for a faster and perhaps more stable enrichment[103]; however, without additional inoculations, their retention may be ultimately subject to environmental changes and other variables that can influence the coral-associated microbiome[104–107].

At T4, 5 months after the last inoculation, the bacterial community shifted back towards pre-inoculation bacterial profiles, which aligns with previous findings in corals[14] and other organisms where the effect of probiotics ceases once the probiotic administration is suspended[108]. Hence, this data provides evidence of such probiotic effect in situ in coral-associated bacteria, evidencing that frequent inoculations can temporarily trigger microbiome restructuring. The necessary frequency of inoculations (days, weeks, or months) and probiotic cell concentration may depend on different variables, such as the goals of the intervention and environmental conditions, and need further investigation and optimization, aiming at reducing

**Fig. 7 | Coral health proxies monitored in *Pocillopora verrucosa*. a** In situ photosynthetic efficiency ($F_v/F_m$) profiles by treatment (placebo and probiotic) and sampling times (T1–T4). The X-axis represents the treatments and the Y-axis shows the $F_v/F_m$ values. $F_v/F_m$ values below 0.6 indicate potentially stressed or damaged photochemical systems of the algae symbionts in corals. **b** ED50 (thermal threshold) (Y-axis) for each treatment (placebo and probiotic) by sampling time (T1–T4) in CBASS experiments. The depicted boxplots show the median (center line) and the first and third quartiles (lower and upper bounds). Biological replicates per sampling time and treatment (*n* = 15).

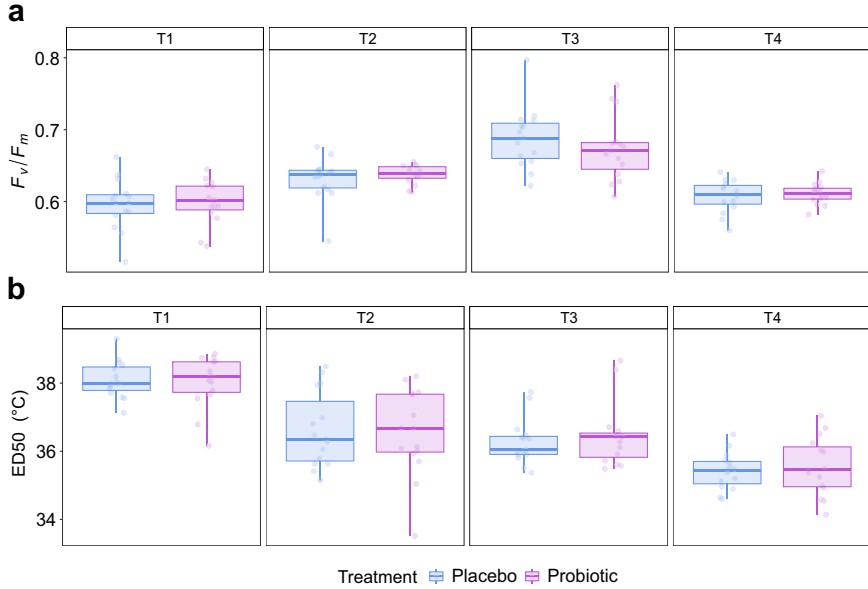

logistics efforts when applying coral probiotics at larger rehabilitation scales. Other factors such as BMCs consortium composition, coral species, multi-host vs single-host bacterial donors, host-microbiome flexibility, and host health (i.e., bleaching state, diseases) likely influence host microbiome-probiotic interactions[10,11]. The potential mechanisms underlying probiotic action in the coral microbiome are likely multi-factorial[53], and may include the enrichment of beneficial bacteria, indirect niche colonization by inoculated BMCs, antagonistic effects against pathogens[13], predatory bacteria[109], immune stimulation by allochthone strains triggered by the probiotic[14,110], and support for coral heterotrophic feeding[36]. Further studies are required to better understand these mechanisms and improve microbiome rehabilitation. Unraveling genomic-level mechanisms associated with BMCs and using omics to track the actual changes promoted by probiotics are crucial steps for addressing knowledge gaps on the functioning of coral probiotics[53], as recently investigated[14,19,40], and would improve the selection of BMCs based on targeted microbial traits.

We assessed the in situ photosynthetic efficiency of *Symbiodiniaceae* and thermo-tolerance response in CBASS experiments as indicators of coral holobiont health. Interestingly, we observed that the ED50s of the investigated coral species changes significantly through time, by almost 2 °C. Additional surveys expanding the number of sampling points and coral species investigated are needed to elucidate the seasonality of ED50 values. Despite these natural variations, we did not observe any significant changes between treatments in the tested health indicators at any of the sampling times. This indicates that the probiotic treatment had no effect on the coral's rapid thermal shock response, as assessed in CBASS experiments. The rapid response to a thermal shock might be influenced by the overall health status of the host, and the time to acclimate to the stress. Healthy microbiomes are more difficult to change[52] and even a restructured healthy microbiome might not quickly acclimate to a rapid thermal shock. The potential probiotic-promoted protection against thermal stress might be rather progressive, and evident in tank experiments, where significant changes in the health status of probiotic-treated corals are only observed when corals are stressed rather than between healthy corals[13,14]. Our results are still valid to confirm the lack of harm caused by the probiotic inoculation on healthy colonies of *P. verrucosa* in situ, which is an urgent risk assessment step that can contribute for science-based frameworks for the safe use of probiotics for wildlife[19]. More importantly, even if there was nothing to be fixed in the coral health and the microbiome was stable, continuous inoculations provide a microbiome (and, potentially, epigenomic[37]) restructuring that could be beneficial in times of stress, requiring further testing and validation. In addition, the observed variations in $F_v/F_m$ at different sampling times do not

appear to be influenced by the inoculation, given that the placebo samples exhibit a similar trend. Hence, the significant temporal differences seem to be more closely connected to the corals' seasonal responses to temperature changes rather than to the experimental treatment. It is a well-established fact that $F_v/F_m$ ratios fluctuate with the seasons, in correlation with alterations in light and temperature conditions, as documented in the literature[111,112]. Notably, the $F_v/F_m$ values for both the placebo and the probiotic-treated corals are indicative of healthy corals. In a nutshell, additional research is required to elucidate the potential protective effect of in situ applied probiotics on the coral's health, in response to rapid and gradual thermal stress.

Further, in situ experiments should investigate the effects of these inoculations in the event of a bleaching event or disease outbreak, and/or the effects of probiotics on diseased or thermally stressed corals displaying signs of bleaching, which would provide insights into the effects of coral probiotics in coral hosts with various health statuses.

The significant changes observed in the coral microbiome and lack of significant effects on the bacterial communities associated with seawater and sediments suggest a targeted effect of coral probiotics, and the absence of impacts on surrounding bacterial communities, for the time periods assayed. In addition to the long-term/permanent effects assessed here, further studies to elucidate the potential short-term effects over these microbial communities are required. The seawater bacterial community displayed more variability over time, while the sediment bacterial community remained stable across sampling times. Coral reefs harbor heterogeneous microbial communities in different niches within the ecosystem[113–118]. It is widely known that the surrounding microbiomes are distinct from the coral microbiome and exhibit different bacterial community profiles with differential functionality[50,119–124]. In the Red Sea, the marked seasonality and environmental drivers influence the dynamics of marine bacterioplankton[125] which likely explains the temporal variation we observed in the seawater bacterial community in the times assayed. Moreover, the sediment bacterial community serves as powerful bioindicators of environmental perturbations in coral reefs[126,127]. The observed stability in the sediment bacterial community reinforces the lack of off-targeted effects in the environment surrounding probiotic-treated corals, even after an intense inoculation period. Additionally, it will be beneficial to further expand the analysis of additional sampling points and other off-targeted organisms including other invertebrates (e.g., sponges) and vertebrates (e.g., fish), that may interact with the inoculated probiotics.

Our findings provide the first evidence that pBMCs can temporarily restructure the coral microbiome of healthy corals in situ, after repeated

inoculations, suggesting their potential incorporation and/or enrichment. Our results also indicate the lack of detectable changes in the surrounding coral microbiomes, providing supporting evidence of the potentially safe application of coral probiotics in reef ecosystems. The risk assessment and studies on the feasibility and logistics required for larger in situ probiotic interventions should be continuously addressed with the upscaling of this rehabilitation strategy. Future studies would also benefit from elucidating the short-term effects of coral probiotics and expanding this research to potential off-target organisms, which would provide an even more comprehensive assessment of the implications and ecological interactions associated with coral probiotic applications in the marine environment at scale. In addition, the protective role promoted by probiotics still needs to be tested in stressed corals in situ.

## Materials and methods
### Bacteria isolation from healthy corals
Coral fragments from *S. pistillata* Clade IV, *G. fascicularis*, and *P. verrucosa*, were collected by snorkeling and Scuba diving at Thala reef (22°15′46.9″ N 39°03′05.9″ E), Aquarium (22°23′15.6″ N 38°55′07.2″ E), and Al Fahal reef (22°18′18.4″ N 38°57′52.5″ E) respectively, in the central Red Sea, at depths of 1–10 m, between February and May 2021. Coral fragments were collected by Scuba diving using gloves and pliers, and transported in 50-ml conical tubes on ice for approximately 1 h to the laboratory. Immediately on arrival, the fragments were macerated using 1–2 mL of 3.5% sterile saline solution with a sterile mortar and pestle. Serial dilutions up to $10^{-6}$ were performed using the macerated paste with 3.5% saline solution, and 100 μL of each dilution was plated in Marine Agar (MA) (Zobell 2216, HiMedia Laboratories, Mumbai, India), adjusted to 3.5% salinity, diluted MA (DMA) (MA medium 2× diluted to 3.5% NaCl), and Luria Bertani agar (LB) (Sigma-Aldrich®), adjusted at 3.5% salinity. Plates were incubated at 25 °C (corresponding to the in situ water temperature registered at the sampling sites at the moment of fragments collection) overnight. In parallel, the coral macerate was incubated in 50 mL of 3.5% saline solution in a 250 mL sterile Erlenmeyer at 27 °C overnight with glass beads at 130 rpm. After this first incubation, triplicate subsamples (100 μL) of $10^{-4}$, $10^{-5}$, and $10^{-6}$ dilutions were plated into MA and DMA culture media and incubated under the same conditions described above. Additionally, 0.5 cm coral fragments were placed on the Petri dishes containing MA and DMA. All the plates were incubated at 25 °C for at least 48 h or until visible bacterial colonies were observed. Approximately 350 bacterial isolates were obtained, based on colony morphology, and were preserved at −80 °C using sterilized glycerol with a final concentration of 20%, for further analysis.

### Bacterial genomic DNA extraction and 16S rRNA gene sequencing of bacterial isolates
Each bacterial isolate from the glycerol stocks was re-grown using 200 μL of the stock and inoculated into 6 mL of Marine broth (HiMedia Laboratories, Mumbai, India), and incubated overnight at 26 °C with 140 rpm agitation. For bacterial DNA extraction, 2 mL of bacterial liquid culture was centrifuged for 5 min at 10,000 rpm to obtain a pellet and washed twice with 3.5% saline solution to wash the cells from the culture media. The DNA extraction was performed using the Wizard® Genomic DNA purification kit (Promega Corporation, USA), following the protocol for gram-positive and gram-negative bacteria. Genomic DNA was purified with the GFX™ PCR DNA and Gel band purification Kit (Cytiva Company, USA) and then quantified using Nanodrop™ 8000 Spectrophotometer (Thermo Scientific™) and Qubit™ dsDNA broad-range assay kit (Invitrogen™). To target the full 16S rRNA gene, universal primers 27F 5'AGAGTTTGATCMTGGCTCAG 3', and 1492R 5' GGTTACCTTGTTACGACTT 3' were used[128], using the AmpliTaq Gold® 360 Master Mix (applied Biosystems®, by Life Technologies™) under the following PCR conditions: one cycle of initial denaturation at 95 °C for 5 min, 30 cycles of denaturation at 95 °C for 1 min, annealing at 50 °C for 1 min, extension at 72 °C for 1 min, and one cycle of final extension at 72 °C for 7 min. The amplification was verified using 1% agarose gels (100 V, 40 min) and visualized in a Bio-Rad® transilluminator.

PCR products were sent to Macrogen (Korea) for taxonomic identification by Sanger sequencing. The forward and reverse sequences (1000–1500 bp) were processed to remove low-quality bases and generate contigs using the ChromasPro software. Ambiguities in the assembled sequences were resolved visually (either by choosing the base from the read with the cleaner signal or changing the consensus base to "*N*"). Cleaned assembled DNA sequences from each of the pBMC isolates were then identified using the EzBioCloud server[129]. The top-hit taxon, obtained from average nucleotide identity, was used to estimate the taxonomy of the pBMC isolates.

### Functional screening of bacterial isolates
Bacterial strains identified as potential human or coral pathogens (e.g., *Vibrio* spp.) were excluded (*n* = 305). The remaining bacterial isolates were tested for beneficial functions, following previous studies[13,14]. For ROS scavengers, 20 μL of pure culture of each bacterial strain was placed on a portable microscope slide, and a drop of 3% (*v/v*) of hydrogen peroxide was immediately added in the center. The criterion of a positive result was evaluated qualitatively, based on the production of bubbles, as a proxy of catalase reaction. Phosphate assimilation was tested according to Nautiyal, 1999[130] using Pikovskaya's agar culture media (HIMEDIA®): 20 μL of pure culture of each strain was dispensed onto the media plate. As bacterial growth occurred overnight, strains that were positive for phosphate assimilation produced a transparent halo around the cultures. Siderophore production was confirmed by plating the isolates on an R2A media plate which contained a cromoasurol and FeCl3 CAS solution. The isolates that could degrade the blue-colored CAS in the media exhibited a yellow-colored halo and thus were regarded as positive[131]. The antagonistic effect against *Vibrio coralliilyticus* (a well-known coral pathogen)[132] was assessed through the diffusion agar method[133]: first, 20 μL of each pBMC bacterial strain was spot-inoculated onto 2.5% NaCl LB agar, placing three spots for each strain (representing replicates). The plates were incubated at 26 °C for as long as necessary to allow the strain to grow. The strains were inactivated by chloroform volatilization, followed by pouring 3 mL of semisolid 2.5% NaCl LB medium (0.7% agar) containing the strain *V. corallilyticus* BAA-450 indicators over the inactivated spots. These plates were incubated at 28 °C for 16 h, and the antagonistic activity was indicated by inhibition halos around or no detection of *Vibrio* growth over the colony spot. Lastly, to evaluate the ability of the strains to hydrolyze urea, we conducted a urease test using Christensen's Urea Agar, as outlined in Brink, 2010 [134]. A droplet from a well-populated, overnight culture of each strain was carefully deposited onto the slanted Urea agar and incubated at 35 °C for a maximum duration of 6 days. During this interval, a positive result was identified by the observable change in the agar's color to a distinct pink hue.

### Selection of pBMCs and probiotic preparation
Six bacterial strains (two *P. galatheae* and two *C. amphilecti* isolated from *P. verrucosa*; one *Halomonas* sp. isolated from *S. pistillata* and one *Sutcliffiella* sp. isolated from *G. fascicularis*) were chosen for the probiotic consortium, based on their potentially beneficial traits (Supplementary Table 1). As the probiotic consortium is composed of a diverse combination of bacteria, each strain was collected proportionally at the peak of the exponential growth phase. Fresh overnight bacterial cultures were collected and washed three times using saline solution (3.5% NaCl) by centrifuging at 6000 g for 5 min each time. Each bacterial strain was resuspended in 100 mL of sterile saline solution (3.5% NaCl). The six strain suspensions were standardized based on colony-forming units, and then they were mixed obtaining a formulation of $10^{8-9}$ cells/mL for the final probiotic consortium.

### Experimental design
The study site was located in a shallow sheltered area in "Al Fahal Reef" (22°18′18.4″ N; 38°57′52.5″ E), a mid-shore reef in the central Red Sea, 15 km off-shore from King Abdullah University of Science and Technology (KAUST), Saudi Arabia (Fig. 1b). The experiment was performed at "the Red Sea Research Center Coral Probiotic Village", a multidisciplinary research initiative established to test the use of coral probiotics and other

pioneering strategies in situ in real coral reef setups. It covers an area of about 500 m², with a maximum depth ranging from 8 to 10 m. In the study area, 30 visually healthy colonies (no visual signs of bleaching or disease) of the brown morphotype of *P. verrucosa* were selected for the experiment (with a minimum distance of 3 m between colonies to minimize sampling of clonal genotypes), and were randomly assigned to the probiotic and control (referred to here as placebo) treatments (*n* = 15 colonies per treatment). The health status of colonies was qualitatively evaluated throughout the experiment, using a coral health chart to assess signs of bleaching. The study was performed from summer to winter 2021 and late spring 2022 to encompass seasonal variations. Four sampling points were considered for analysis: T1, before the treatment inoculations (late August 2021); T2, after one and a half months of inoculations (mid-October 2021); T3, at the end of the inoculations (late November 2021); and T4, 5 months after the last inoculations (April 2022). Inoculations were performed repeatedly with a frequency of three times per week during a 3-month period (T1–T3), using 50 ml plastic syringes containing 30 mL of the probiotic consortium (with a final concentration of $1 \times 10^{8-9}$ cells/mL), released slowly over the coral colony (Supplementary Fig. 3). The placebo treatment consisted of an autoclaved 3.5% NaCl solution (the same used to resuspend pBMC-cells) applied in the same way. The use of inert negative controls (i.e., without the addition of any confounding factors), is the gold standard procedure for testing probiotics, as described in ref.[53]. Dead cells should not be used as a negative control as they can also trigger specific responses in the inoculated hosts and are, therefore, not inert[53]. Fragments from each colony were collected for coral-associated bacterial community analysis by Scuba diving at all sampling times (T1–T4), and before any probiotic inoculations at the moment of sampling, using sterile gloves and pliers (one for each treatment) and individual sterile collection bags (Whirl-Pak®). On the boat, immediately after collection, coral fragments were placed in sterile 5 mL cryovials and covered with DESS buffer (20% dimethyl sulfoxide, 0.25 M ethylenediaminetetraacetic acid, and saturated sodium chloride (NaCl), with adjusted pH 8.0), and immediately snap-frozen in liquid nitrogen. Samples were transported to the laboratory (less than 3 h after collection) and stored at −80 °C until further processing. In parallel, sediments and water surrounding the coral colonies were collected at T1 and T3 to monitor their bacterial communities and assess water nutrients and dissolved organic carbon. The surrounding water and sediments from 10 of the studied *P. verrucosa* colonies distributed in different areas of the experimental study site (*n* = 5 per treatment) were sampled (using a random number generator): sediment samples were collected at the bottom of each colony between 1 and 5 cm depth approximately, using sterile 50 mL falcon tubes. Water samples were collected approximately 30 cm distant from the same colonies, using 2 L dark bottles that were acid-washed in HCl 4% for 10 min prior to the sample collection. Samples were stored on ice on the boat and filtered the same day upon arrival in the laboratory, using a filtration rack with 0.22 µm Millipore Sigma membranes attached to a vacuum pump. Filter membranes were individually stored at −80 °C, for less than two weeks, until DNA extraction. All equipment and materials used were thoroughly sterilized to avoid contamination. All sampling procedures were carried out within a one-week interval for each sampling time (T1–T4). During T1–T3, samples of coral, seawater, and sediment were collected for microbial community analysis on the same day, immediately before starting the placebo and probiotic inoculations.

## Monitoring of in situ physicochemical parameters and inorganic nutrients

Seawater temperature and salinity were monitored throughout the duration of the experiment using multiparameter CTDs (Ocean Seven 310 Multiparameter CTD, Idronaut). The daily minimum, maximum, and mean seawater temperature, and mean salinity values are summarized in Supplementary Table 8. Seawater collected during T1 and T3 was used for inorganic nutrient analysis of the surrounding water of each of the randomly selected coral colonies (*n* = 5 per treatment). Briefly, water was filtered on the boat, with 0.22 µM Millex®-GV filters (PVDF Membrane, Merck Millipore Ltd.,

Ireland), into 15 mL falcon tubes. Subsequently, samples were placed on ice on the boat and then frozen at −20 °C until analysis. The inorganic nutrients analyzed were: Silica ($Si(OH)_4$), Nitrite ($NO_{2-}$), Nitrate ($NO_{3-}$), and Phosphate ($PO_4^{3-}$). All measurements were performed using a segmented flow analyzer (Model AA3 HR, SEAL Analytical Inc.) with the following detection limits: Silicates 0.08322 µmol L⁻¹; Nitrite 0.0217 µmol L⁻¹; Nitrate 0.0322 µmol L⁻¹; and Phosphate 0.01052 µmol L⁻¹ (Supplementary Table 8).

## In situ photosynthetic efficiency for coral health monitoring

The photosynthetic efficiency of the algae symbionts (*Symbiodiniaceae*) was assessed through the maximum quantum yield of PSII photochemistry $F_v/F_m$. A pulse-amplitude modulation (PAM) diving-PAM system (Diving PAM II, Walz) with a red-emitting diode was used (LED; peak at 655 nm). PAM data was collected after sunset, at least 30 min after complete darkness, to ensure there was full photochemical dissipation of the reaction centers. The diving PAM was configured as follows: measuring light intensity = 6; gain = 2; and damping = 4.

The changes in photosynthetic efficiency ($F_v/F_m$) over time in different treatments were analyzed using a linear mixed effect model using the function "lmer" from R package *lme4*[135] in R studio (R Core Team). Colony (biological replicates, *n* = 15) nested to treatment was treated as a random effect on the intercept to account for the non-independence of replicates with time. $F_v/F_m$ was included in the model as a response variable, sampling time as a predictor variable, and treatment as a factor with two levels: probiotic and placebo. We performed model selection using likelihood-ratio tests starting with the most complex model and sequentially removing terms until all parameters were significant at $p < 0.05$. Changes in the $F_v/F_m$ over time were tested using the "emmeans" R function in all pairwise combinations.

## CBASS experiments

To evaluate coral heat response behavior during the experiment, short-term acute heat stress assays were performed to determine the coral thermal threshold at the genotype (per colony) level. Here, CBASS was used as a proxy to assess coral health and determine if the long-term inoculation of coral probiotics had an effect on coral thermal tolerance threshold, as well as their natural thermotolerance variation during a seasonal time frame. In sampling times T1, T2, T3, and T4, four fragments of each colony were collected. The fragments were transported in seawater to the wet lab facility of the Coastal and Marine Resources Core Lab (CMR, KAUST), where the set-up was ready to receive the corals. Briefly, the system consists of four 10 L flow-throughs supplied with raw seawater collected from the site a day before the runs. Each tank runs different temperature regimes independently, and the light setting was adjusted to correspond to in situ irradiance (600 µmol photons m − 2 s − 1), which was adjusted using an LI-193 Spherical Underwater Quantum Sensor (LI-COR) and manual adjustment of dimmable 165 W full spectrum LED aquarium lights (Galaxyhydro). The lights followed a 12:12 h day/night cycle. The temperature of each tank was controlled using the ITC-310T-B (Inkbird) thermostat connected to an IceProbe Thermoelectric chiller (Nova Tec) and 200 W titanium aquarium heaters (Schego). HOBO Pendant® Temperature Data Loggers (Model UA-001-64) recorded the temperature of each tank every 10 min during the experiment. One fragment corresponding to each colony was exposed to a different temperature condition. The temperature regime of each tank was 1 control/baseline: 30 °C, 1 medium: 33 °C, 1 high: 36 °C, and 1 extreme: 39 °C. The CBASS assays ran for 18 h, where the temperature of the 30 °C tank was maintained at 30 °C for the entire experiment; in the other tanks, the temperature was increased to 33 °C, 36 °C, and 39 °C, respectively, and then returned to 30 °C overnight until the end of the experiment. The detailed temperature profiles are provided in the supplementary material (Supplementary Table 9). After 7 h from the start of the experiment (and 1 h in darkness), we measured the endosymbiotic algae photosynthetic efficiency ($F_v/F_m$) for all fragments using a PAM fluorometry (Diving PAM II, Walz). The measurement also matched the temperature ramping down to 30 °C.

The data were analyzed according to Voolstra and collaborators (2020)[51], where $F_v/F_m$ values were used to evaluate the treatment's ED50, corresponding to its thermal threshold. ED50 corresponds to the effective doses that cause a 50% decrease in the $F_v/F_m$. The dose-response curves were fitted using the R package "drc"[136]. The changes in ED50 with sampling time at different treatments were analyzed using a linear mixed effect model using the function lmer from R package "lme4"[135] in R studio (R Core Team). Colony (biological replicates, $n = 15$) nested to treatment was treated as a random effect on the intercept to account for the non-independence of replicates with time. $F_v/F_m$ was included in the model as a response variable, sampling time as a predictor variable, and treatment as a factor with two levels: probiotic and placebo. Model selection was performed using likelihood-ratio tests starting with the most complex model and sequentially removing terms until all parameters were significant at $p < 0.05$. Changes in the ED50 over time were tested using the "emmeans" R function in all pairwise combinations.

## DNA extraction, library preparation, and sequencing of bacterial communities

The DNA from the coral fragments was extracted using a DNeasy® Blood & Tissue kit (Qiagen) according to manufacturer instructions, with the gram-positive bacteria pre-treatment and the following modification: coral fragments of approximately 0.5 g were used directly for the extraction. The lysis incubation step after adding proteinase K was carried out overnight for approximately 16 h at 56 °C, with constant agitation at 650 rpm in a Thermomixer (ThermoFisher®).

DNA was extracted from water samples using a DNeasy® Blood & Tissue kit (Qiagen) by cutting the filter into small pieces with a sterile cutter and tweezers. The protocol was performed following manufacturer instructions, with the following modifications: at the sample pre-extraction preparation stage, half of the membrane filter was cut into smaller pieces and then placed in 1.5 mL microcentrifuge tubes. The volume of all the following solutions used in the kit was adjusted to similar proportional volumes (thus not changing any concentration of compounds) to fully immerse all the membrane filter pieces into the solution: 540 μL ATL buffer and 60 μL Proteinase K were added. After adding these solutions, the incubation step at 56 °C was conducted for 3 h. Then, the volumes of buffer AL and ethanol were 400 μL, buffer AW1, and AW2 were 500 μL, and the final elution buffer AE was 50 μL. DNA samples were stored at −20 °C until downstream analyses.

DNA was extracted from sediment samples using a DNeasy® PowerSoil Kit (Qiagen), with the following modification: 12.5 μL of Proteinase K was added to approximately 0.5 g of sediments for incubation overnight at 56 °C, with constant agitation at 650 rpm in a Thermomixer. The downstream steps were performed according to the kit's protocol. DNA concentration and purity for all samples (coral, water, and sediment) were quantified using a Qubit™ dsDNA assay kit (Invitrogen™) and Nanodrop™ 8000 Spectrophotometer (Thermo Scientific™). Sequencing of the V3–V4 regions of the 16S rRNA gene was performed using the universal primers 341F 5′ CCTACGGGNGGC WGCAG 3′ and 785R 5′ GAC TAC HVG GGT ATC TAA TCC 3′ for the coral, sediment, and water samples, at Novogene Corporation-Inc in China. In brief, PCR mixtures contained 15 μL of Phusion® High-Fidelity PCR Master Mix (New England Biolabs), 0.2 μM of each of the forward and reverse primers, and 10 ng of the samples' genomic DNA. The thermal cycling conditions were as follows: a first denaturation step at 98 °C for 1 min, followed by 30 cycles at 98 °C for 10 s, 50 °C for 30 s, and 72 °C for 30 s, and a final extension of 5 min at 72 °C. PCR products were verified and quantified by mixing their equal volume with 1× loading buffer (contained SYB green) and performing electrophoresis on 2% agarose gels. For the library preparation, PCR products were purified using a Qiagen Gel Extraction Kit (Qiagen, Germany). Sequencing libraries were generated with a NEBNext® Ultra™ II DNA Library Prep Kit (Cat No. E7645). The library quality was evaluated on a Qubit@ 2.0 Fluorometer (Thermo Scientific™) and Agilent Bioanalyzer 2100 system. Libraries were sequenced on a NovaSeq platform (Illumina) and 250 bp paired-end reads were generated.

All sequence reads were deposited in the European Nucleotide Archive (ENA) under the study accession number PRJEB65896.

## Bacterial community analyses

The *DADA2* pipeline was used to infer amplicon sequence variants (ASVs)[137] using the 16S rRNA gene-based amplicon libraries of coral, sediment, and water. Briefly, the raw reads were decontaminated of phiX, and adapter-trimmed using the "BBDuk" tool from the BBMap suite (Bushnell B, http://sourceforge.net/projects/bbmap/). PCR primers were then removed from the reads using the "cutadapt" tool[138]. After performing concatenation of the forward and reverse reads via "justConcatenate" option in the "mergePairs" function of *DADA2*, the sequences were analyzed under the pseudo-pooling mode by following the standard *DADA2* (version 1.22) workflow and using the SILVA database, version 138.1[139]. The potential contaminant ASVs that were identified in the negative controls and the study samples were removed from the analysis by the "decontam" tool[140] using the prevalence-based method (on the default threshold setting).

In brief, reads corresponding to mitochondria, chloroplast, archaea, eukaryotes, and singletons were removed, resulting in 46,803 ASVs for the coral dataset, 34,099 for the water dataset, and 68,354 for the sediment dataset. Alpha and beta diversity, plots, ordinations, and statistical comparisons were carried out in R version 4.2.2 (R Core Team, 2018) using the functions in *Phyloseq* version 1.42.0[141] and *Vegan* version 2.6-4[142]. All plots were generated using *ggplot2* version 3.4.0. Additional figures to represent the experimental design were created in a licensed version of BioRender. Alpha diversity of the coral bacterial community was calculated using the rarefied ASV counts to the minimum sample depth (18,364 reads) with the function "estimate. diversity" from *Phyloseq*, with the default diversity indices (Observed S, Shannon H', Simpson, and Chao1). Statistical comparisons between treatments for alpha diversity metrics were calculated by implementing the Wilcoxon test, and previous testing of the null hypothesis for normal distribution of the data (Shapiro–Wilks, $p$ value < 0.05). nMDS analyses were generated from Bray–Curtis distances of Wisconsin-square root transformed ASV total counts using the "vegdist" and "metaMDS" functions in *Vegan* for the coral dataset. Principal component analysis was implemented for the water and sediment datasets from Bray–Curtis distances. Statistical differences in the microbial communities of coral, seawater, and sediment components were assessed using the transformed ASV counts to relative abundance. The permutational multivariate ANOVA test (PERMANOVA) was implemented to test for significance, using "sampling time" and "treatment" as factors, implementing the "adonis2" function in *Vegan*, from generated Bray–Curtis distances and 999 permutations. The homogeneity of variances was calculated between treatments (placebo and probiotic) using the "betadisper" and "permutest" functions in *Vegan* using Bray–Curtis distances and 999 permutations for the coral, water, and sediment datasets. For the coral dataset, this was calculated for each sampling time. In T3, the variances between treatments in the coral dataset were not homogeneous (Betadisperse, d$f = 1$, $F = 7.2107$, $Pr (>F)$ 0.012). Nonetheless, as PERMANOVA is largely unaffected by heterogeneity in balanced designs[143], we proceeded to calculate the statistical significance of the treatment using the *Adonis* function for T3 samples (biological replicates, placebo: $n = 14$; probiotic: $n = 15$).

Comparisons to evaluate changes in the relative abundance of the dominant bacterial families in the coral microbiome between treatments (placebo and probiotic) in T3 were carried out using the two-sided Wilcoxon test, after testing for normal distribution of the data (Shapiro–Wilk, $p$ values < 0.05). The enrichment of the ASVs corresponding to the pBMCs genera in the coral microbiome in T3, was assessed by comparing their relative abundance between treatments, using the two-sided Wilcoxon test, and previous testing for normal distribution of the data (Shapiro–Wilk, $p$ values < 0.05). The genus *Sutcliffella* was not detected in the coral dataset; nonetheless, this genus was previously part of the *Bacillus* genus and was recently re-classified[100]. The coral 16S rRNA gene amplicon data might not

reflect its most recent taxonomy; therefore, we included the genus *Bacillus* as a proxy of *Sutclifiella* in these comparisons. In addition, as the 16S amplicon data used in the study does not provide taxonomic resolution to the species level, the ASV sequences were queried against the 16S sequences for each of the six pBMCs used, using BLASTn[144] to identify them in the coral microbiome. From the original dataset (previously removing singletons in Phyloseq), we identified seven ASVs that had 100% match across the entire amplicon to the V3–V4 region of the 16S rRNA gene in one of the pBMCs, and only four (corresponding to pBMC *Cobetia amphillecti* and pBMC *Halomonas* sp.) were retained after singletons were removed (Supplementary Table 10). The relative abundance of those ASVs was extremely low (<0.1%) in the coral microbiome, and they were not detected at sampling time T3.

To identify differentially abundant ASVs between placebo and probiotic treatments in the coral microbiome in T3, the analysis of the composition of microbiomes with bias correction "ANCOM-BC2"[145] was used. This method estimates unknown sampling fractions, corrects bias from sample differences, models absolute abundance with linear regression, and provides a statistically valid test with appropriate $p$ values, false discovery rate control, and sustained power. We performed this analysis on total ASV counts (after removing singletons), using the Benjamini–Hochberg method to correct for false positives and an alpha of 0.05 for significance. An ASV was considered significant when it was enriched or decreased significantly ($p$-adj. < 0.05) in the probiotic samples in comparison to the placebo (reference group) under the aforementioned parameters. We focused on the top 20 most enriched ($p$-adj. < 0.01, $W$ statistic > 4) and 20 most decreased ($p$-adj. < 0.01, $W$ statistic < 4) ASVs.

### Reporting summary

Further information on research design is available in the Nature Portfolio Reporting Summary linked to this article.

## Data availability

All sequence reads were deposited in the ENA under the study accession number PRJEB65896. Other data supporting the results of this study are provided as Supplementary information files. Large data sets are available on Zenodo repository https://doi.org/10.5281/zenodo.10801800.

## Code availability

All R code used in this study is available in the Zenodo repository https://doi.org/10.5281/zenodo.10801800.

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

## Acknowledgements

We acknowledge KAUST Core Labs and CMOR staff for their technical and logistics support for laboratory processing and diving operations. This work was supported by KAUST grant number BAS/1/1095-01-01 and the KAUST Center Competitive Funding (CCF) FCC/1/1973-51-01. Eslam Osman is partially funded by a postdoctoral fellowship provided by Ocean Science and Solutions Applied Research Institute (OSSARI), Education, Research, and Innovation (ERI) Sector, NEOM, Tabuk, Saudi Arabia.

## Author contributions

Study conception: Peixoto, R.S., Villela, H. and Garcias-Bonet, N. Experimental design: Peixoto, R.S., Villela, H., Garcias-Bonet, N., Delgadillo-Ordóñez, N., Voolstra, C.R., Benzoni, F., Carvalho, S. and Berumen, M.L. Coral identification: Benzoni, F. Bacteria isolation, and probiotic preparation: Delgadillo-Ordóñez, N., Raimundo, I., Villela, H., Alsaggaf, A., Cardoso, P., Barno, A., and Peixoto, R.S. Acquisition of field data: Delgadillo-Ordóñez, N., Raimundo, I., Villela, H., Garcias-Bonet, N., Osman, E.O., Cardoso, P., Barno, A., Santoro, E.P., Rosado, J.G.D., García, F.C., and Peixoto, R.S. Laboratory processing: Delgadillo-Ordóñez, N., Raimundo, I., Santoro, E.P., García, F.C., Cardoso, P., Garcias-Bonet, N., Barno, A. and Rosado, J.G.D. Molecular work: Delgadillo-Ordóñez, N., Raimundo, I., and Bocanegra, C. Data processing: Antony, C.P. Formal analyses: Delgadillo-Ordóñez, N., García, F.C., Garcias-Bonet, N., and Curdia, J. Interpretation of data and discussions: Delgadillo-Ordóñez, N., García, F.C., Garcias-Bonet, N., Curdia, J., Voolstra, C.R., Carvalho, S., and Peixoto, R.S. Drafting: Delgadillo-Ordóñez, N., Garcias-Bonet, N., García, F.C. and Peixoto, R.S. Critical revision by all authors.

## Competing interests

The authors declare no competing interests.
