## [Peer Review File · Communications Biology]

Reviewers' comments:

Reviewer #1 (Remarks to the Author):

The manuscript from Delgadillo-Ordoñez et al. describes a fascinating proof of concept in situ experiment of probiotic treatment of corals. The authors treated *Pocillopora verrucosa* colonies in situ in the Red Sea with a mixture of potentially beneficial bacterial isolates. The authors show convincingly that this continuous treatment with the bacteria for three months influence the coral microbiome into a potentially beneficial direction. In addition, the authors analyzed the surrounding seawater and sediment, which seemed to be unaffected.

My main criticism relates to the statement that probiotics are incorporated into the microbiome of the treated corals (line 217). For this conclusion I miss the underlying data support. If this is true, then I expect to find the same 16S rRNA gene sequence (100% sequence similarity) of the inoculated bacteria in the microbiome data of the coral. However, the authors identified these bacteria only on the genera level without defining the sequence similarity. Here, I encourage the authors to compare the sequences of inoculated and colonizing bacteria based on 100% sequence similarity.

Specific points:

Line 80 /Table S1. Please describe, why these traits are considered as beneficial?

Line 106: Please also consider the effect size by adding the R2 values as a measure of effect size to all statistical tests.

Line 112: Please define BMC-phylotypes? are these the same ASVs with 100% identity between inoculated and colonizing bacteria?

Figure 3 B. For consistency, please use the same scale for relative abundance in Figure 3B and 4A, 0-1 or 0-100.

Figure 5: Are the ASVs representing the inoculated bacteria shown in this heatmap? If yes, please mark them.

Figure 6: To conclude that the inoculated bacteria are not found in the surrounding environment, please analyze the water and sediment samples also on ASV level.

Line 217: I miss the data support for this main conclusion.

Line 266: The authors should bear in mind that not all *Vibrio* are pathogenic. Similar to the argument with *Endozoicomonas*, there is a whole range between mutualistic and pathogenic *Vibrio*. Please be more specific here.

Line 275. Please define, what you mean by "BMC genera" on the level of sequence similarity.

Line 414. How did the authors ensure an equal number of each isolate in the inoculum? How did you estimate the cell numbers, by counting? Can you please add the information of cells/ml at OD600=0,1.

Line 437. Here, it is not clear, if this is the final total number of bacterial cells, or of each bacterial strain?

Reviewer #2 (Remarks to the Author):

The authors of this publication have put together an important study looking at the effects of BMCs (coral probiotics) in an in situ setting. Their results suggest that some of the BMC strains incorporate into the coral microbiomes while some apparently do not, however, they do not appear to affect the surrounding environment.

I believe this to be an important study that provides very important information for the field as a whole. This type of work has not been published before and would be important for subsequent studies and initiatives for the BMC field as a whole. However, because of this, I felt there were a few points that should be addressed that would vastly strengthen this study. If these could potentially be addressed, I believe it would make it a very strong and important publication. My major points are detailed below.

- Lines 76-79 (In general references these strains as BMCs): I suggest rewording this study a bit. Technically, these strains have not been demonstrated to be probiotics in the lab, if I'm not mistaken. Unless they were tested for their ability to mitigate thermal stress in another study? Therefore, I don't think they can be referred to as probiotics/BMCs demonstrating that. I would be fine, if this was just phrased as a proof-of-concept for in situ bacterial inoculation. I don't think this refutes the study, because underwater inoculation with any non-pathogenic bacteria on corals have not been published and that would be a great proof-of-concept, but I don't think these strains can be referred to as BMCs at this point.

- In regards to using microbiome sequencing: I agree that the monitoring of the coral microbiome is great, however, it would make sense to also check for the presence of the inoculated strains using qPCR. I believe the authors could see shifts in the microbiomes, but it could be that the resolution of this approach was not high enough to detect specific strains. I recommend that qPCR primers be developed for the inoculated strains and that the samples should be screened. This would be fairly straightforward and would strengthen this study if the DNA extractions were kept.

- Lines 112 - 115: IN regards to the shifts in *Halomonas* and *Pseudoalteromonas*. Again, this is a great finding, however, I think the conclusions could be strengthened by the qualification of the inoculated bacteria using something like qPCR. Further, these great result could be made even more compelling if, for example, the *Pseudoalteromonas* and *Halomonas* ASVs identified in the microbiome sequencing could be compared directly to the sequences from the inoculated strains and the percent similarities

reported somewhere?

-Lines 115 - 116: In regards to the *Cobetia* and *Sutcliffiella* not being detected in the microbiome studies. It is possible that these genera are not detected using microbiome sequencing, but may be present via more targeted approaches, e.g., PCR. Maybe not, but a simple endpoint PCR with specific primers for the strains (with Sanger sequencing of some of the product) might give an answer. E.g., endpoint PCR that target a unique intergenic region. If there's a product then you can purify it, clone it into a replicative plasmid, transform into *E. coli*, and sequence some of them to see if they match the original strain. If there's no PCR product, then your job is done and it strengthens your conclusion.

-Lines 186 - 192: In regards to the testing of in situ environmental samples. I do appreciate the testing of this, but for testing this important safety aspect for the use of the probiotics in situ, shouldn't this have been checked at more time points? What if there were shorter term changes? I only bring this up because it seems to be a major focus on the study but it's only been checked at two time points. What if their effects are not based on time? For example, water flow, or temperature? I feel a bit uncomfortable if the emphasis of this paper is put on this finding (and implying safety) when the corals were sampled at 4 time points and these were only done at 2 time points. I realize the difficulties with environmental sampling, so if these samples just don't exist then I would maybe take some of the emphasis off of the effects on the environment or phrase is that the effects were not observed in the time frames tested and maybe make the title less of a conclusive statement. For example, the "in situ effects were investigated" instead of a conclusion.

-Lines 206-208: In regards to significant changes to the Fv/Fm rates after inoculation. Isn't this a little concerning? The temperature was lower at T4 versus T1, so it doesn't point to thermal stress. Can there be other metrics or metadata provided to account from these differences? From this statement, one could infer that the inoculation of these corals could have a significant effect on the endosymbiont functionality, which contradicts the previous statement. I may be interpreting these statement incorrectly, if so, can this section be reworded?

Minor Comments:

-Lines 225-228: It could be pointed out that any benefits from *Endozoicomonadaceae* have not actually been proven. So their loss cannot technically be attributed as negative or positive at this point.

-Lines 238-241: *Rhodobacteraceae*, also includes members enriched in diseased corals. I would suggest not overemphasizing their benefits.

-Line 242-255: This section seems redundant with the previous section.

Reviewer #3 (Remarks to the Author):

This paper reports on a field study of coral probiotics. The authors conducted a robust experiment and I

applaud their effort. However, there are key problems with the interpretation of results, particularly regarding (1) the effort required to yield a response, (2) the loss of an effect within months of stopping treatments, and (3) the importance of a lack of effect on coral thermal tolerance (despite all evidence suggesting an effect should be present).

30 corals on the reef are tagged, and 15 are given probiotic treatments while the other 15 are given placebo treatments at an interval of 3-time-per-week for 3 months. At four timepoints, the corals are surveyed/sampled for the in situ photochemical efficiency of their symbionts, microbiome analysis, and thermal tolerance assays. The study reports no effect on the environment as the major result of this paper. However, there are a number of more important results that should be discussed in more detail and should feature in the main result and in the abstract. Given that water is moving past corals on the reef continuously, it is not surprising that the experimental scale treatments have not affected the environment. However, the really important results here (that are touched upon too briefly in its current form) are that: (1) alteration of microbiome in the field takes an immense effort (3 time-per week for 3 months only yields a response after 3 months), (2) There is no impact on thermal tolerance, despite laboratory evidence that this treatment should have an effect, and (3) That the ED50 of individual colonies are highly variable through time (an immensely important result).

My suggestion is that the study needs to bring the thermal tolerance results into the main paper, not as supplementary materials, and put a stronger focus (throughout the paper, including the abstract) on the effort limitations of probiotic treatments as a management (which to me is the main result of the paper) rather than the fact that the treatments did not affect the environment (which is not that surprising).

Major Comments

Major Comment 1

The effect of 3-time-per-week treatments is only measurable after takes 3 months.

This is an enormous effort, to then lose the effect within 5 months of stopping treatments. Therefore, considerable effort would likely be required to implement this strategy as a rehabilitation/conservation strategy. Additionally, the forecasted onset of marine heatwaves is not accurate for a 3-month outlook. Therefore, the conclusion in the discussion (L290-295) that suggests it would be worth applying these probiotic treatments to rehabilitate corals under NOAA bleaching Alerts fails to recognise the limitations to the accuracy of the outlook products, and the mismatch between time needed to generate an effect of probiotics. Even if the field application of probiotics could increase tolerance (which it did not in this study), then the treatment would need to be started before the weather forecasts could even predict that a heatwave was likely. As such this proactive approach would be needed each year, in anticipation that heatwave might occur, as the time for a microbiome response to take effect is longer in duration than the time for a heatwave to be forecasted. In practice, applying such treatments to large numbers of corals on a reef and even over multiple locations would be logistically extremely challenging and costly. Therefore, this absolutely must be clear in both the abstract and introduction and should not be buried in the methods. Again, this highlights the incredible effort needed to conduct a probiotic rehabilitation project in practice. A more thoughtful message is needed in this paper to highlight these limitations.

Major Comment 2

The study found no impact of probiotic treatments on coral thermal tolerance (ED50s), despite success of these strains in altering thermal tolerance in lab studies (all BMCs are antagonists of *Vibrio coralliilyticus* – a pathogen known to compromise thermal tolerance, Yael Ben-Haim 2003 – and BMCs increase catalase and reduce urease – so should have elicited a thermal tolerance response). This finding is highly important, and should be a main part of the paper, not buried in supplementary information. The Figure S3 should be in the main manuscript and should be a major focus of discussion of the study (included in the abstract).

Major comment 3

This study conducted CBASS experiments on the same 30 colonies over an 8-month period. This is a great achievement. The study clearly shows that the ED50s change significantly through time, by almost 2C. That is a huge result, as it helps us to understand how flexible the CBASS response is – a potentially major limitation to determining coral thermal tolerance. Again, this should be a main result. The only other explanation is that there are issues with the relevance of the rapid thermal shock response of a CBASS experiment in capturing a relevant measure of thermal tolerance. If that is the case, it should be explored in the discussion.

Major comment 4

There is no mention in the methods of when samples were collected (for CBASS, in situ Fv/Fm, and microbiome) in relation to when the colonies were inoculated with probiotic treatments. Sampling just after inoculation would of course be a big problem. If that is not the case, there is no problem, it should just be stated. If that is the case, then a clear discussion of this limitation is required. For instance, could the result that there was only an effect at T3 be due to sampling just after the probiotic treatment was applied? This must be clarified

In-line comments

L21-23: This is an overstatement. While the authors are talking about only the “treatments”, there are a whole host of other mechanisms that reef managers use to ensure that native corals are not lost. Here, retain, may also be the wrong word to use. Consider alternative to state: “... can enhance the stress resistance of corals in laboratory trials and is now being considered as a treatment for stressed corals in the wild.”

L23: should read as “To elucidate this, we inoculated *Pocillopora verrucosa* colonies at a Red Sea reef with BMC treatment three times per week for a three-month period”

L24: The intensity of treatment application must be provided in the abstract. Otherwise, it sounds like a one-off treatment may be enough, and risks overselling the result and being blown out of proportion by the media etc. that will only read the abstract.

L25: “appeared” this choice of wording does not inspire confidence in the result. If there was a robust experimental design, and there was statistically no change, then state that the microbiome of surround areas “remained” unchanged.

L28-29: Makes it sound like this was circumstantial. If it correlated with decreased vibrio, state that more clearly: "... and a decrease in potential coral pathogens, such as Vibrio."

L30-32: This statement doesn't make sense as it should say what exactly the data indicate about feasibility. Notably, given that the microbiome effect was gone after a few months, given that it took 3 months to achieve an effect, and given that there was no impact on thermal tolerance, I think the main message of this abstract should highlight the limitations of probiotics and state the need for more research.

L36: "to society"

L39: Neither of these citations are independently peer reviewed. One is an editorially reviewed report and the other is report. Should cite the primary literature for this statement.

L41: should say "enhancing the natural resilience of corals to cope"

L50: rapid adaptation (i.e., increase in frequency of heat tolerance related genes in a population) occurs through genetic means, over multiple generations. Please either state how changing symbiotic partners can help corals adapt, or reword the sentence.

L66: This is where the "three times per week" should be stated. For example: "In this study, we inoculated colonies of the scleractinian *Pocillopora verrucosa* (Ellis and Solander, 1786) with probiotic treatments three times per week over a three-month period to investigate ..."

L69: should read as "an overall restructuring of the coral microbiome"

Figure 1 C – very hard to read dates, consider just showing the 1st of each month in a simple format, rather than the entire dates. Still can keep the minor ticks as weekly,

Figure 2 – Show statistical results here too (only significant difference at T3) and show the percentage variance explained by different MDS axes.

Figure 3 – were corrections for multiple testing conducted?

L193: should say "yet remained stable over time" as it is in comparison to the previous sentence, given the usage of "similarly" in this sentence.

L214 missing a close bracket

L273-274: This study showed completely the opposite – that the new community did not impact thermal stress tolerance at all.

290-295 – This is an interpretation beyond what the results show. In fact, the results show that inoculating corals with probiotics that improve stress tolerance in the lab do not improve stress

tolerance in the field. Therefore, the advice should be the opposite – that rehabilitation projects should be very careful before relying solely on probiotics – as it shows a limited capacity for success in a field trial. This type of over-reaching should not be included

L308: Here the authors should state the message of Major Comment 1 above.

L321: The assumption at the beginning of a heat stress assay should be that all corals included are healthy (unless there are hypotheses about prior health on thermal performance – not the case in this study). Therefore, this is not a reason to discount the results of CBASS experiments.

L324: Suggest that probiotics do not seem to cause harm: surely the hope is that probiotics would be beneficial?

L326: The CBASS experiment is a 'time of stress', and the results show no effect.

L446: should now state when samples were collected in relation to when the probiotic treatment was applied. It is a problem if the probiotic treatment is applied just before the microbiome samples was taken.

Title: **PROBIOTICS RESHAPE THE CORAL MICROBIOME *IN SITU* WITHOUT DETECTABLE PERMANENT OFF-TARGETED EFFECTS IN THE SURROUNDING ENVIRONMENT**

Reviewer: 1

The manuscript from Delgadillo-Ordoñez et al. describes a fascinating proof of concept *in situ* experiment of probiotic treatment of corals. The authors treated *Pocillopora verrucosa* colonies *in situ* in the Red Sea with a mixture of potentially beneficial bacterial isolates. The authors show convincingly that this continuous treatment with the bacteria for three months influence the coral microbiome into a potentially beneficial direction. In addition, the authors analyzed the surrounding seawater and sediment, which seemed to be unaffected.

My main criticism relates to the statement that probiotics are incorporated into the microbiome of the treated corals (line 217). For this conclusion, I miss the underlying data support. If this is true, then I expect to find the same 16S rRNA gene sequence (100% sequence similarity) of the inoculated bacteria in the microbiome data of the coral. However, the authors identified these bacteria only on the genera level without defining the sequence similarity. Here, I encourage the authors to compare the sequences of inoculated and colonizing bacteria based on 100% sequence similarity.

A: We thank you for your constructive comments and feedback to our manuscript. We believe our article truly benefited from them. We have addressed your concern regarding the potential incorporation of the bacterial strains used in the probiotic consortium into the coral microbiome, and now the manuscript addresses this aspect with clarity. We identified the ASVs in the coral microbiome matching 100% similarity, with the inoculated strains. The respective ASVs were found either absent or in extremely low abundance in the 16S amplicon data, despite the significant enrichment of the bacteria corresponding to the same genera of some of the inoculated pBMCs. We addressed this limitation in the discussion (Lines 243-258). Please also note that this conundrum does not indicate ‘lack of success’: even in the absence of long-term enrichment of the probiotic strains at high abundance, microbial transfer can still trigger a ‘reboot’ or ‘reset’ of the microbiome that lead to a more beneficial microbiome (as reviewed in Garcias-Bonet et al., 2023). Thus, the restructuring of the microbiome following probiotic treatment, in addition to (ideally), finding the treated bacterial strains in the restructured microbiome are both indicators of a successful treatment. We re-phrased our statement (Lines 243-258) as follows, for more precision in our conclusions:

“Here we demonstrated that pBMCs can instigate a restructuring of the stable microbiome of healthy corals *in situ* without causing permanent changes in the microbial communities of the surrounding environment. Continuous probiotic application led to shifts in the taxonomic composition and diversity of the healthy (and therefore robust⁵²) microbiome of *P. verrucosa*, which has also been previously shown to be very resistant to changes in their microbiome⁵⁰. Still, the continued inoculations promoted an enrichment of the bacteria corresponding to the same genera of some of the inoculated pBMCs. This may suggest that probiotics can be incorporated by corals *in situ* and/or trigger bacterial enrichment in the coral microbiome. Although the inoculated genera were enriched and we identified the ASVs in the coral microbiome matching 100% similarity with the inoculated pBMCs, those ASVs were eventually found absent or in extremely low abundance in the 16S amplicon data. This does not indicate “lack of success”: even in the absence of long-term manifestation of the probiotic strains at high abundance, microbial transfer can still trigger a “reboot” or “reset” of the microbiome that leads to

a potentially more beneficial microbiome (as reviewed in Garcias-Bonet et al., 2023⁵³). Thus, the restructuring of the microbiome following probiotic treatment, in addition to (ideally) finding the treated bacterial strains in the restructured microbiome, are both indicators of a successful treatment.”

Line 80 /Table S1. Please describe, why these traits are considered as beneficial?

A: Thank you for pointing this out. We added and described the potentially beneficial traits tested *in vitro*, as well as the positive test for each selected pBMC, as follows (Lines 82-95):

“Putative BMC strains were selected based on exhibiting at least one of the following assumed beneficial traits via *in vitro* testing: Antagonistic effect against the coral pathogen *Vibrio corallilyticus* (measured through the diffusion agar method)^{43,44}, reactive Oxygen Species (ROS) scavenging (measured through catalase activity), which potentially minimizes ROS concentration during thermal stress¹⁰; production of siderophores (measured through siderophores excretion), which bind to iron compounds and increase their concentration, making them into bioavailable forms for supporting Symbiodinaceae metabolism^{45,46}; phosphate assimilation (measured through positive activity of phosphate-solubilizing bacteria) to support the coral metabolism^{2,47}; and urease activity, (measured through urease secretion to hydrolyze urea), to support nitrogen cycling by making bioavailable nitrogen compounds for the coral holobiont^{48,49}. From the six selected pBMCs strains, the two *Pseudoalteromonas galathea* (30H & 31H) were positive for catalase, *Cobetia amphilecti* strains (65H & 81H) were positive for catalase and phosphate assimilation, *Halomonas* sp. (SAT10) was positive for catalase and siderophores production, and *Sutcliffiella* sp. was positive for catalase and siderophores production. (Table S1).”

R: Line 106: Please also consider the effect size by adding the R2 values as a measure of effect size to all statistical tests.

A: We have added the R2 values (to indicate effect size) to each statistical test, accordingly.

R: Line 112: Please define BMC-phylotypes? are these the same ASVs with 100% identity between inoculated and colonizing bacteria?

A: We agree the use of phylotype was not clear enough and we have re-phrased as follows to avoid any misunderstanding on the concept (Lines 127-128):

“We also detected an enrichment in ASVs belonging to the same genera of the inoculated pBMCs (Fig. 3B).”

R: Figure 3 B. For consistency, please use the same scale for relative abundance in Figure 3B and 4A, 0-1 or 0-100.

A: Thank you for this suggestion. As the genera represented in Figure 3B were found in very low abundance in the coral microbiome (<0.1% of the total microbiome), according to the 16S rRNA sequencing data, changing the scale from 0-100 would make the boxplots unreadable.

R: Figure 5: Are the ASVs representing the inoculated bacteria shown in this heatmap? If yes, please mark them.

A: The exact ASVs representing the inoculated bacteria are not shown in the heatmap, as the represented ASVs correspond to the most significant ASVs according to the differential abundance analysis ANCOM-BC2.

R: Figure 6: To conclude that the inoculated bacteria are not found in the surrounding environment, please analyze the water and sediment samples also on ASV level.

A: Thank you for your suggestion. To clarify on this aspect, we did not intend to probe that the inoculated bacteria are not present in the seawater and sediment surrounding the colony, as these bacteria can naturally inhabit the reef ecosystem and the water column. What we intend to probe here is that the repeated inoculations of the probiotic consortium did not change the bacterial community structure of the seawater and sediment, when comparing at two sampling points (T1 and T3). This is clarified in the results section (Lines 203-208), as follows:

“In contrast to the changes observed in corals, the bacterial communities associated with the surrounding seawater and sediment were not affected by the probiotic inoculation when comparing samples collected near coral colonies under different treatments before (T1) and after (T3) the probiotic inoculation period (see details in methods). As the pBMCs can naturally occur in the reef environment, we focused on the overall effect of the treatment in the bacterial community structure of seawater and sediment.”

R: Line 217: I miss the data support for this main conclusion

A: We re-phrased our statement and we believe that now it clearly reflects the main findings of this work (Lines 243-258): “Here we demonstrated that pBMCs can instigate a restructuring of the stable microbiome of healthy corals *in situ* without causing permanent changes in the microbial communities of the surrounding environment. Continuous probiotic application led to shifts in the taxonomic composition and diversity of the healthy (and therefore robust⁵²) microbiome of *P. verrucosa*, which has also been previously shown to be very resistant to changes in their microbiome⁵⁰. Still, the continued inoculations promoted an enrichment of the bacteria corresponding to the same genera of some of the inoculated pBMCs. This may suggest that probiotics can be incorporated by corals *in situ* and/or trigger bacterial enrichment in the coral microbiome. Although the inoculated genera were enriched and we identified the ASVs in the coral microbiome matching 100% similarity with the inoculated pBMCs, those ASVs were eventually found absent or in extremely low abundance in the 16S amplicon data. This does not indicate “lack of success”: even in the absence of long-term manifestation of the probiotic strains at high abundance, microbial transfer can still trigger a “reboot” or “reset” of the microbiome that leads to a potentially more beneficial microbiome (as reviewed in Garcias-Bonet et al., 2023⁵³). Thus, the restructuring of the microbiome following probiotic treatment, in addition to (ideally) finding the treated bacterial strains in the restructured microbiome, are both indicators of a successful treatment.”

Line 266: The authors should bear in mind that not all *Vibrio* are pathogenic. Similar to the argument with *Endozoicomonas*, there is a whole range between mutualistic and pathogenic *Vibrio*. Please be more specific here.

R: We agree the reduction in members of Vibrionaceae bacteria may encompass pathogenic and non-pathogenic bacteria. Therefore, we re-phrased this paragraph as follows (Lines 303-311):

“Moreover, we observed a decrease in bacteria from the family Vibrionaceae, which encompasses a wide range of marine bacteria, extensively associated to coral microbiomes⁶⁶. Some Vibrionaceae members constitute opportunistic coral pathogens, likely contributing to coral diseases and bleaching^{43,44,92–99}. We observed that the

probiotic treatment led to a decrease in the abundance of *Vibrio* spp., Unclassified Vibrionaceae, *Photobacterium*, and *Catenococcus*. Previous studies have shown that probiotics can reduce *Vibrio* abundance and mitigate coral bleaching^{13,14}. Nonetheless, it is important to note that some *Vibrios* can be non-pathogenic⁶⁹, and therefore, further studies exploring the restructuring of Vibrionaceae with targeted approaches would help enlighten specific shifts in pathogenic and non-pathogenic coral-associated bacteria.”

R: Line 275. Please define, what you mean by “BMC genera” on the level of sequence similarity.

A: We modified the text to clarify the term “BMC genera”, referring to the ASVs in the coral microbiome that were identified at the genus level to the same genera used in the probiotic inoculations (Lines 312-313).

“We also explored the aforementioned presence and relative abundance of the ASVs corresponding to the same genera of the inoculated pBMCs in the coral microbiome.”

R: Line 414. How did the authors ensure an equal number of each isolate in the inoculum? How did you estimate the cell numbers, by counting? Can you please add the information of cells/ml at OD600=0,1.

A: We included the requested information to obtain similar concentrations of each bacterial strain for the probiotic consortium (Lines 487-493):

“As the probiotic consortium is composed of a diverse combination of bacteria, each strain was collected proportionally at the peak of the exponential growth phase. Fresh overnight bacterial cultures were collected and washed three times using saline solution (3.5% NaCl) by centrifuging at 6000 g for 5 min each time. Each bacterial strain was resuspended in 100 mL of sterile saline solution (3.5% NaCl). The six strains suspensions were standardized based on Colony Forming Units (CFU), and then they were mixed obtaining a formulation of 10⁸⁻⁹ cells/mL for the final probiotic consortium.”

R: Line 437. Here, it is not clear, if this is the final total number of bacterial cells, or of each bacterial strain?

A: Thanks for pointing this out, we have now improved clarity of the text, as detailed in the previous question (just above). The new text clarifies the protocol used to ensure a similar concentration of each strain composing the probiotic consortium.

Reviewer: 2

The authors of this publication have put together an important study looking at the effects of BMCs (coral probiotics) in an in situ setting. Their results suggest that some of the BMC strains incorporate into the coral microbiomes while some apparently do not, however, they do not appear to affect the surrounding environment.

I believe this to be an important study that provides very important information for the field as a whole. This type of work has not been published before and would be important for subsequent studies and initiatives for the BMC field as a whole. However, because of this, I felt there were a few points that should be addressed that would vastly strengthen this study. If these could potentially be addressed, I believe it would make it a very strong and important publication. My major points are detailed below.

R: Thank you for your positive and constructive feedback on our work. We believe your comments and suggestions benefited our manuscript and improved clarity for the specific aspects mentioned, also avoiding any misleading statements. We are happy to have addressed all your comments, which have strengthened our manuscript.

R: Lines 76-79 (In general references these strains as BMCs): I suggest rewording this study a bit. Technically, these strains have not been demonstrated to be probiotics in the lab, if I'm not mistaken. Unless they were tested for their ability to mitigate thermal stress in another study? Therefore, I don't think they can be referred to as probiotics/BMCs demonstrating that. I would be fine, if this was just phrased as a proof-of-concept for in situ bacterial inoculation. I don't think this refutes the study, because underwater inoculation with any non-pathogenic bacteria on corals have not been published and that would be a great proof-of-concept, but I don't think these strains can be referred to as BMCs at this point.

A: Thank you for your critical review. You are absolutely right, apologies for the miswording. We re-phrased our manuscript referring to the bacterial strains used as putative probiotic bacteria (pBMCs), as this is the first study testing their potential to reshape the coral microbiome *in situ*, not to promote their health (as they were already healthy). These strains were also analyzed in a separate study for their whole genomes (Raimundo et al., in prep.) to screen additional putative beneficial pathways to the coral holobiont, which will provide further insights into the metabolic potential of these pBMCs. We would like to mention that our lab is carrying out several experiments in parallel (using other colonies, more coral species and different experimental designs) that include testing the probiotic consortium in aquaria setups and in the field, to test for their beneficial effect upon challenging the corals to different stressors (e.g. coral pathogens and elevated temperature), and during this year's bleaching event, in the Red Sea, and that these results indicate their beneficial role, although we agree this is still the initial step on the study of these probiotics, and we still need to treat them as pBMC.

R: - In regards to using microbiome sequencing: I agree that the monitoring of the coral microbiome is great; however, it would make sense to also check for the presence of the inoculated strains using qPCR. I believe the authors could see shifts in the microbiomes, but it could be that the resolution of this approach was not high enough to detect specific strains. I recommend that qPCR primers be developed for the inoculated strains and that the samples should be screened. This would be fairly straightforward and would strengthen this study if the DNA extractions were kept.

A: We agree that using qPCR to test for the presence of the specific inoculated strains in the coral samples is a more sensitive approach to detect their presence. Indeed, we designed specific primers for each inoculated bacterium and performed qPCR trials (data not shown). However, the concentration of the target sequences seems to be below the qPCR detection limit for environmental samples, preventing their detection and quantification in our coral samples. Other studies in our lab have successfully detected and quantified these strains by qPCR in mock communities, composed by known enriched numbers of the same bacterial isolates (Alsaggaf et al., *In prep*), which confirms that although qPCR is a more sensitive method, it is still not enough to detect these strains in complex samples, which are mostly dominated by host DNA.

R: - Lines 112 - 115: IN regards to the shifts in *Halomonas* and *Pseudoalteromonas*. Again, this is a great finding, however, I think the conclusions could be strengthened by the qualification of the inoculated bacteria using something like qPCR. Further, these great result could be made even more compelling if, for example, the

Pseudoalteromonas and Halomonas ASVs identified in the microbiome sequencing could be compared directly to the sequences from the inoculated strains and the percent similarities reported somewhere?

A: This is a very good suggestion, also shared by R1 and incorporated in the new version of the manuscript. We identified the ASVs in the coral microbiome matching 100% similarity, with the inoculated strains (Table S10). The respective ASVs were found either absent or in extreme low abundance in the 16S amplicon data. We addressed this limitation in the discussion (Lines 250-258). We still see, however, the significant enrichment of the exact same genera significantly correlated with probiotic inoculation. Please also note that this conundrum does not indicate 'lack of success': even in the absence of long-term manifestation of the probiotic strains at high abundance, microbial transfer can still trigger a 'reboot' or 'reset' of the microbiome that lead to a potentially more beneficial microbiome (Garcias-Bonet et al., 2023). Thus, the restructuring of the microbiome following probiotic treatment, in addition to (ideally) finding the treated bacterial strains in the restructured microbiome are both indicators of a successful treatment. We re-phrased our statement (Lines 250-258) as follows, for more precision in our conclusions:

“Although the inoculated genera were enriched and we identified the ASVs in the coral microbiome matching 100% similarity with the inoculated pBMCs, those ASVs were eventually found absent or in extremely low abundance in the 16S amplicon data. This does not indicate “lack of success”: even in the absence of long-term manifestation of the probiotic strains at high abundance, microbial transfer can still trigger a “reboot” or “reset” of the microbiome that leads to a potentially more beneficial microbiome (as reviewed in Garcias-Bonet et al., 2023⁵³). Thus, the restructuring of the microbiome following probiotic treatment, in addition to (ideally) finding the treated bacterial strains in the restructured microbiome, are both indicators of a successful treatment.”

R: Lines 115 - 116: In regards to the Cobetia and Sutcliffiella not being detected in the microbiome studies. It is possible that these genera are not detected using microbiome sequencing, but may be present via more targeted approaches, e.g., PCR. Maybe not, but a simple endpoint PCR with specific primers for the strains (with Sanger sequencing of some of the product) might give an answer. E.g., endpoint PCR that target a unique intergenic region. If there's a product then you can purify it, clone it into a replicative plasmid, transform into E. coli, and sequence some of them to see if they match the original strain. If there's no PCR product, then your job is done and it strengthens your conclusion.

A: Thank you for your suggestion. To address this point, we performed qPCR with specific primers designed to detect the inoculated strains in the coral microbiome samples. However, although qPCR is a more sensitive method, it is still not enough to detect these strains in complex samples, which are mostly dominated by host DNA. We provided clarification on this aspect in a previous question (above).

R: Lines 186 - 192: In regards to the testing of in situ environmental samples. I do appreciate the testing of this, but for testing this important safety aspect for the use of the probiotics in situ, shouldn't this have been checked at more time points? What if there were shorter term changes? I only bring this up because it seems to be a major focus on the study but it's only been checked at two time points. What if their effects are not based on time? For example, water flow, or temperature? I feel a bit uncomfortable if the emphasis of this paper is put on this finding (and implying safety) when the corals were sampled at 4 time points and these were only done at 2 time points. I realize the difficulties with environmental sampling, so if these samples just don't exist then I would maybe take some of the emphasis off of the effects on the environment or phrase is that the effects were not observed in the

time frames tested and maybe make the title less of a conclusive statement. For example, the "in situ effects were investigated" instead of a conclusion.

A: Thank you for your input, this is a great point. Although we sampled corals and surrounding samples at these specific sampling points, the inoculation of probiotics was continuous. That means, probiotics were inoculated three days before samples were collected, in all sampling points. In this sense, we somehow acknowledged both long-term and short-term changes, as we can detect the cumulative effects (comparing sampling points), but also any short-term changes (comparing different treatments at the same sampling point), as probiotics were continuously applied - although no changes were detected. *It also refers to the contrast of observing changes in the coral microbiome at the same sampling point where no changes can be detected in the surrounding samples.* We also expected the microbiome associated with the surrounding sediments to be more susceptible to environmental changes, and, perhaps respond to continuous and frequent inoculation of probiotics that could correlate with permanent changes, as marine sediment microbial communities are good bioindicators of environmental perturbation (see Rodriguez et al., 2021; Glas et al., 2019). Such good indicators of environmental perturbation are not indicating significant impacts either. While we acknowledge the significance of assessing additional sampling points, which may offer even finer time-resolution insights into concerns related to immediate changes or impacts from environmental factors (e.g., temperature, water flow), *our study focuses on the broader context of detecting major and permanent changes after repeated inoculations.* As this field experiment required a massive and time and resource-consuming effort (which included field work/diving three times a week, every week, for three months to inoculate the corals), the collection of additional samples were not logistically feasible for this particular experiment. Still, we present new, urgent and valuable evidence demonstrating that this long-term inoculation of probiotics do not permanently disrupt the natural microbial communities in surrounding seawater and sediment after a three-month inoculation period. Rather than focusing on small, time-sensitive bacterial community changes, our emphasis is on providing a first baseline evidence supporting the putative safety of probiotic inoculation over the long term (and therefore the field effort focus was on inoculations rather than sampling points). As the pioneering study on *in situ* manipulation of putative Beneficial Microorganisms (pBMCs) in wild corals, we believe establishing this first safety baseline is crucial.

To further address this valid concern, we have refined our language to underscore the need for additional studies to explore the short-term effects of coral probiotics in the reef, encompassing other organisms such as invertebrates (sponges) and vertebrates (fish). Ongoing research from our lab includes two separate studies investigating the effects of in situ application of probiotics in fish and sponges, along with their associated microbial communities (Rosado & Delgadillo, *in prep*; Ribeiro et al., *in prep*), and even longer-term (two years) inoculations and several additional sampling points (Garcia et al., ongoing experiments). We changed the title of the manuscript and also addressed this comment in the discussion (Lines 382-400), as follows:

“The significant changes observed in the coral microbiome and lack of significant effects on the bacterial communities associated with seawater and sediments suggest a targeted effect of coral probiotics and the absence of impacts on the surrounding environment, for the time periods assayed. In addition to the long-term/permanent effects assessed here, further studies to elucidate potential short-term effects over these microbial communities are required. The seawater bacterial community displayed more variability over time, while the sediment bacterial community remained stable across sampling times. Coral reefs harbor heterogeneous microbial communities in different niches within the ecosystem^{113–118}. It is widely known that the surrounding microbiomes are distinct from the coral microbiome and exhibit different bacterial community profiles with differential functionality^{50,119–124}. In the Red Sea, the marked seasonality and environmental drivers influence the dynamics of marine

bacterioplankton¹²⁵ which likely explains the temporal variation we observed in the seawater bacterial community in the times assayed. Moreover, the sediment bacterial community serve as powerful bioindicators of environmental perturbations in coral reefs^{126,127}. The observed stability in the sediment bacterial community reinforces the lack of off-targeted effects in the environment surrounding probiotic-treated corals, even after an intense inoculation period. Additionally, it will be beneficial to further expand the analysis of additional sampling points and other off-targeted organisms including other invertebrates (e.g. sponges) and vertebrates (e.g. fish) that may interact with the inoculated probiotics.”

R: Lines 206-208: In regards to significant changes to the Fv/Fm rates after inoculation. Isn't this a little concerning? The temperature was lower at T4 versus T1, so it doesn't point to thermal stress. Can there be other metrics or metadata provided to account from these differences? From this statement, one could infer that the inoculation of these corals could have a significant effect on the endosymbiont functionality, which contradicts the previous statement. I may be interpreting these statement incorrectly, if so, can this section be reworded?

A: The observed variations in Fv/Fm at different sampling times do not appear to be influenced by the inoculation, given that the control samples exhibit a similar trend. We didn't observe a treatment effect in Fv/Fm. Hence, the significant temporal differences seem to be more closely connected to the corals' seasonal responses to temperature changes rather than to the experimental treatment. It is a well-established fact that Fv/Fm ratios fluctuate with the seasons, in correlation with alterations in light and temperature conditions, as documented in the literature (Scheufen et al., 2017; Aichelman et al., 2019; García et al., under review). Notably, the Fv/Fm values for both the control and the probiotic-treated corals are indicative of healthy corals. Throughout the experiment, the corals maintained their health, which aligns with the lack of observable treatment-related differences in Fv/Fm – which is expected, as probiotics are usually effective in dysbiotic/unhealthy microbiomes. In case of occurrences of coral paling or lower Fv/Fm values, then a treatment effect would have been expected. We clarified this aspect as follows (Lines 370-377):

“In addition, the observed variations in Fv/Fm at different sampling times do not appear to be influenced by the inoculation, given that the control samples exhibit a similar trend. Hence, the significant temporal differences seem to be more closely connected to the corals' seasonal responses to temperature changes rather than to the experimental treatment. It is a well-established fact that Fv/Fm ratios fluctuate with the seasons, in correlation with alterations in light and temperature conditions, as documented in the literature^{111,112}. Notably, the Fv/Fm values for both the control and the probiotic-treated corals are indicative of healthy corals.”

Minor comments

R: Lines 225-228: It could be pointed out that any benefits from Endozoicomonadaceae have not actually been proven. So their loss cannot technically be attributed as negative or positive at this point.

A: We agree. Endozoicomonadaceae cannot be classified as beneficial bacteria for corals, as recent studies proposed dynamic roles within the coral holobiont (Neave et al., 2017; Pogoreutz et al., 2018; Tandon et al., 2022). We also believe, although we have not investigated/have no data to support it, that a better taxonomic resolution of currently sequences assigned as *Endozoicomonas* may elucidate this issue in the future. We emphasized your point as follows (Lines: 266-271).

“As the beneficial role of *Endozoicomonas* is yet to be proved, and we did not observe any negative effect on the coral health upon the probiotic inoculations (discussed later), we argue that the decrease in Endozoicomonadaceae did not reflect in a detrimental effect on the coral holobiont, and may have led to an enrichment of other key bacterial groups (detailed hereafter) that could increase the holobiont resilience in the event of environmental impacts.”

R: Lines 238-241: Rhodobacteraceae, also includes members enriched in diseased corals. I would suggest not overemphasizing their benefits.

A: We re-phrased this sentence to provide a broader view of Rhodobacteraceae and its roles in the coral holobiont, as follows (Lines 277-281):

“Rhodobacteraceae is a common member of coral microbiomes correlated with various health statuses ^{61,66,68}. Although their role in the coral holobiont is unclear, they seem to be involved in key functions that can promote coral health, including nitrogen cycling, toxic compound degradation, antimicrobial activity ^{69,70}, being also commonly found associated with the mucus of coral larvae ³⁵.”

R: Line 242-255: This section seems redundant with the previous section.

A: We believe that this section provides further information on the specific bacterial groups that were enriched with the probiotic inoculations, highlighting potential beneficial roles of such groups in the coral holobiont, which are an important focus of the discussion. We have shortened and improved the readability of this section as follows (Lines 281-302):

“Some ASVs significantly enriched in probiotic-treated corals are potentially beneficial for the coral holobiont. Some examples include *Simkania* (Simkaniaceae), a coral endosymbiont occurring in close association with *Endozoicomonas* bacteria ⁷¹; *Delftia* (Commamonadaceae), a key member of coral microbiomes ⁷² that plays roles in anti-quorum sensing and antibiofilm activity ⁷³ and may help to control pathogenic microbes associated with bleaching ⁷⁴; and *Ruegeria* (Rhodobacteraceae), known for their role in antimicrobial effects against coral pathogens ⁷⁵, colonization of early life stages of coral ⁷⁶, and degradation of toxic compounds ⁷⁰. Other examples include fermentative bacteria such as *Limosilactobacillus* (Lactobacillaceae), formerly classified as *Bacillus*, isolated from healthy coral mucus ⁷⁷, capable of forming stable associations with probiotic bacteria from the genus *Lactobacillus* ⁷⁸. Other fermentative bacteria include Rikenellaceae RC9 gut group, *Saccharofermentas*, *Ruminococcus*, *Pseudobutyrvibrio*, *Butyrvibrio*, and Christensenellaceae R-7 group, which may play a key role in carbon metabolism and nitrogen cycling within the coral holobiont ^{63,79,80}. They potentially contribute to the degradation of complex carbohydrates (i.e. starch) produced by Symbiodiniaceae ^{63,81}. Some nitrogen-fixing bacteria were also enriched, including *Rhodococcus* (Nocardiaceae), also known for its antimicrobial activity ^{82,83} and degradation of emergent contaminants in the marine environment ^{84,85}. Other nitrifiers included Mle-1-7 group (Nitrosomonadaceae) and *Nitrospira* (Nitrospiraceae), which may be important for the primary productivity of coral photosynthetic symbionts by making nitrogen compounds available ⁸⁶. We also observed the enrichment of coral intracellular protozoan endosymbionts, such as *Candidatus* Amoebophilus ⁸⁷⁻⁸⁹, which interact with eukaryotic hosts, such as *Symbiodinium* spp. and apicomplexans ^{90,91}.”

Reviewer 3:

This paper reports on a field study of coral probiotics. The authors conducted a robust experiment and I applaud their effort. However, there are key problems with the interpretation of results, particularly regarding (1) the effort required to yield a response, (2) the loss of an effect within months of stopping treatments, and (3) the importance of a lack of effect on coral thermal tolerance (despite all evidence suggesting an effect should be present).

30 corals on the reef are tagged, and 15 are given probiotic treatments while the other 15 are given placebo treatments at an interval of 3-time-per-week for 3 months. At four timepoints, the corals are surveyed/sampled for the in situ photochemical efficiency of their symbionts, microbiome analysis, and thermal tolerance assays. The study reports no effect on the environment as the major result of this paper. However, there are a number of more important results that should be discussed in more detail and should feature in the main result and in the abstract. Given that water is moving past corals on the reef continuously, it is not surprising that the experimental scale treatments have not affected the environment. However, the really important results here (that are touched upon too briefly in its current form) are that: (1) alteration of microbiome in the field takes an immense effort (3 time-per week for 3 months only yields a response after 3 months), (2) There is no impact on thermal tolerance, despite laboratory evidence that this treatment should have an effect, and (3) That the ED50 of individual colonies are highly variable through time (an immensely important result).

My suggestion is that the study needs to bring the thermal tolerance results into the main paper, not as supplementary materials, and put a stronger focus (throughout the paper, including the abstract) on the effort limitations of probiotic treatments as a management (which to me is the main result of the paper) rather than the fact that the treatments did not affect the environment (which is not that surprising).

A: We thank the constructive criticism provided to our work and appreciate your feedback and suggestions to help us provide the most accurate wording that can reflect our scientific data. We are happy to have addressed all your concerns and comments in the corresponding sections. We trust that the manuscript has highly benefited from these insights and rich discussion on several aspects that were mentioned across the revisions.

Overall, we fully agree that it is important to highlight that *this manuscript is the first step of a long scientific road* and is rather focused on the microbiome survey in the face of microbiome manipulation of coral reefs, *in situ*, and the application aspects, although highly relevant, were not tested and therefore cannot be the main topic/focus. The use of probiotics for wildlife is a new area of research to be tackled by several research groups and projects. *Each piece of the puzzle is urgent, and altogether, data from field surveys can define the limitations, impact, improvements and knowledge that can be obtained from these experiments.* Although our group is (and other groups also are) developing a number of parallel studies to elucidate the many variables and questions, the first one of them is addressing the basic knowledge gap: *how do natural and healthy coral colonies, and their surroundings, respond to microbiome manipulation?* This is the focus of this study and our results are the first of many datasets that will contribute to the rapid development of this important field of research. We have edited the text to clarify this. In addition, and more specifically addressing the highlighted questions/observations:

(1) Alteration of microbiome in the field takes an immense effort (3 time-per week for 3 months only yields a response after 3 months):

In this work, yes. This is a good point, and we have made sure to highlight it even more. It is widely known, from other studies, in tanks and other organisms, that probiotics work better when the holobiont is under stress. This is one of the reasons we don't consider probiotics a "solution" or even "one of the solutions" to "save coral reefs", at least as per the scientific data and strategies available. We have repeatedly said that in scientific and media publications. Probiotics can be used, however, as temporary and very welcomed customized medicine (Peixoto, Sweet and Bourne, 2019), rather than a permanent approach, targeting the timing of the applications based on temperature stress forecasts. This strategy could maximize the feasibility of using microbial therapies to reduce mortality and retain native coral colonies of specific reefs at scale while permanent solutions are developed. Although we did not use stressed corals, to know how quick they would incorporate probiotics and if this incorporation would promote their health, this caveat was included in the text.

(2) There is no impact on thermal tolerance, despite laboratory evidence that this treatment should have an effect:

It is true that there was no impact measured, based on the proxies used, which was also highlighted in the current version, but also there was no impact at all. Changes were therefore not "needed", perhaps even possible to be detected by CBASS or any (most?) coral proxies. All corals are healthy, and, *exactly as it is also observed in tank experiments (Santoro et al., 2021; Rosado et al., 2019), it is therefore not possible to detect differences that will surely arise when corals are stressed (which is what is compared in tank experiments).* It is similar to treating a healthy person with antibiotics - if there is nothing to be cured, how will the outcome be evaluated? The point here, however, is that this is a field that is still in its infancy. We are moving as we develop it, and before we (and while we wait for) test it during a heatwave, we wanted to develop this baseline, ecological study tracking the effects of microbiome manipulation *in situ* (in corals and their surrounding environment), which is, independently of its applications, an exciting and urgent scientific question that is addressed in this manuscript.

(3) That the ED50 of individual colonies are highly variable through time (an immensely important result):

This is an excellent point, and we agree. We also noticed this very interesting data and, to confirm, and further explore this data, we combined the ED50 data obtained in this experiment with additional field surveys, including a higher number of colonies and species, which we analyzed and submitted as a short communication that is currently under review. The reviewer is right that this needs to be mentioned here too (although the dataset is not as complete as it is in our parallel study). To quickly jump to the point: several coral species were observed to exhibit seasonal thermal thresholds, i.e. acclimate or adjust their thermal tolerance through time. We are currently preparing another study that is demonstrating exactly this point.

Major Comment 1

R: The effect of 3-time-per-week treatments is only measurable after takes 3 months. This is an enormous effort, to then lose the effect within 5 months of stopping treatments. Therefore, considerable effort would likely be required to implement this strategy as a rehabilitation/conservation strategy. Additionally, the forecasted onset of marine heatwaves is not accurate for a 3-month outlook. Therefore, the conclusion in the discussion (L290-295) that suggests it would be worth applying these probiotic treatments to rehabilitate corals under NOAA bleaching Alerts fails to recognise the limitations to the accuracy of the outlook products, and the mismatch between time needed to generate an effect of probiotics. Even if the field application of probiotics could increase tolerance (which it did not in this study), then the treatment would need to be started before the weather forecasts could even predict that a heatwave was likely. As such this proactive approach would be needed each

year, in anticipation that heatwave might occur, as the time for a microbiome response to take effect is longer in duration than the time for a heatwave to be forecasted. In practice, applying such treatments to large numbers of corals on a reef and even over multiple locations would be logistically extremely challenging and costly. Therefore, this absolutely must be clear in both the abstract and introduction and should not be buried in the methods. Again, this highlights the incredible effort needed to conduct a probiotic rehabilitation project in practice. A more thoughtful message is needed in this paper to highlight these limitations.

A: We acknowledge your comment regarding potential limitations in deploying coral probiotics as a rehabilitation strategy in coral reefs. This is a very useful feedback, and we agree. We have therefore added some missing aspects to either include or elucidate the caveats. For example, the observed temporary effect induced by probiotics in the coral's microbiome, evident after 3 months of inoculations, is expected and aligns with findings indicating temporal effects of probiotics in various organisms, including humans (Garcias-Bonet et al., 2023). The absence of a sustained restructuring effect five months after the last inoculation underscores the transient nature of these microbial changes. The three-month timeframe required to reshape the microbiome of *P. verrucosa* may be attributed to the lack of need, as all corals were healthy and the microbiomes were, therefore, more stable (Anna Karenina Principle). These caveats reinforce an important limitation of the use of probiotics, which, in their current form/application, are only potential emergency band-aids. However, for them to become actual band-aids or even more permanent treatments, many different research projects are needed, in order to detect and define the bottlenecks and how such delivery can be improved. One of the main questions is the feasibility of delivering them (under stress or not) and potential off-target changes in the microbiome. These are the topics we tried to partially address in this manuscript. By recognizing the logistical challenges of *in situ* probiotic deployment, we are also actively engaged in ongoing research to optimize delivery methods in reef environments. This includes exploring options such as autonomous dispensers and probiotic encapsulation in novel materials. These efforts aim to streamline delivery processes, reduce inoculation times, and enhance the targeted effects of probiotics on coral health and physiology. Our lab, alongside many others globally, is dedicated to advancing these research endeavors to contribute to more effective and practical coral reef rehabilitation strategies. We have addressed this aspect through the discussion, as detailed below (Lines 328-331 & 337-343)

“On the contrary, the enrichment of probiotics seems to be facilitated when corals are under stress¹⁴, which may indicate that, if the goal is the rehabilitation and retention of threatened corals, the use of probiotics as medicine applied in times of stress may be a good strategy to be tested^{10,102}.” (Lines 328-331)

“At T4, five months after the last inoculation, the bacterial community shifted back towards pre-inoculation bacterial profiles, which aligns with previous findings in corals¹⁴ and other organisms where the effect of probiotics ceases once the probiotic administration is suspended¹⁰⁸. Hence, this data provides evidence of such probiotic effect *in situ* in coral-associated bacteria, evidencing that frequent inoculations can temporarily trigger microbiome restructuring. The necessary frequency of inoculations (days, weeks, months) and probiotic cell concentration may depend on different variables, such as the goals and environmental conditions, and need further investigation.” (Lines 337-343).

R: Major Comment 2

The study found no impact of probiotic treatments on coral thermal tolerance (ED50s), despite success of these strains in altering thermal tolerance in lab studies (all BMCs are antagonists of *Vibrio coralliilyticus* – a pathogen known to compromise thermal tolerance, Yael Ben-Haim 2003 – and BMCs increase catalase and reduce urease

– so should have elicited a thermal tolerance response). This finding is highly important, and should be a main part of the paper, not buried in supplementary information. The Figure S3 should be in the main manuscript and should be a major focus of discussion of the study (included in the abstract).

A: We appreciate your comments and suggestions and now present in Fig.7 in the main text. We also highlight this finding, although this was not the focus of our study, which was aiming at providing a first microbiological survey exploring coral microbiome manipulation *in situ*. While previous studies have demonstrated significant effects of Beneficial Microorganisms (BMCs) in improving the health of *stressed corals* in aquarium experiments (Santoro et al., 2021; Rosado et al., 2019), two key considerations are: 1, in our study all colonies used, both placebo and probiotic-treated, maintained visually healthy throughout the experiment, showing high Fv/Fm rates and no signs of bleaching or disease; 2, in tank experiments we can also only see significant changes when corals were stressed. Healthy corals (and organisms overall) have robust and stable microbiomes that are hard to change (Anna-Karenina principle), and it is also difficult to promote health to an already very healthy organism (i.e., how to fix something that is not broken?). This is why the focus of this work, conducted on healthy corals, was to further explore if long-term applications would finally restructure the coral microbiome, and whether changes in the surrounding environment would be triggered by this constant probiotic application. We have three other papers in preparation that were conducted by other students and members of the team in different years (2022 and 2023) and levels of heat stress. These new results demonstrate significant health improvements, using different proxies, when corals are actually stressed. In such experiments, probiotics were applied just before (days) or during heat stress and not only restructured the coral microbiome and metabolome within a few weeks but also significantly improved their health. The data from these experiments, conducted on different coral colonies and additional species, are still being processed and constitute another step of development of the field. *This is a new field of research to be developed by many research groups, over many years, and things are being developed as we report them. We agree we still have more questions than answers, but each piece of the puzzle is important (and urgent!) for the scientific community. One of them is what happens when healthy corals and their surrounding are exposed to probiotics, and this is the question we are focused on here.* We agree it is important to highlight all these aspects and indicate the limitations of using probiotics on healthy (or any) corals, and we appreciate the suggestions, which were incorporated to the new version, as follows, through different sections in the abstract (Lines 29-31), and the discussion (Lines: 328-331 & 356-381):

“As all corals (treated and non-treated) remained healthy throughout the experiment, we could not track health improvements or protection against stress.” (Lines 29-31)

“On the contrary, the enrichment of probiotics seems to be facilitated when corals are under stress¹⁴, which may indicate that, if the goal is the rehabilitation and retention of threatened corals, the use of probiotics as medicine applied in times of stress may be a good strategy to be tested^{10,102}.” (Lines 328-331).

“We assessed the *in situ* photosynthetic efficiency of Symbiodiniaceae and thermo-tolerance response in CBASS experiments as indicators of coral holobiont health. Interestingly, we observed that the ED50s of the investigated coral species changes significantly through time, by almost 2 °C. Additional surveys expanding the number of sampling points and coral species investigated are needed to elucidate the seasonality of ED50 values. Despite these natural variations, we did not observe any significant changes between treatments in the tested health indicators at any of the sampling times, likely due to the overall healthy status of the colonies throughout the experiment. Healthy microbiomes are more difficult to change⁵² and the probiotic-promoted protection against stress cannot be tested without stress, *as it has been demonstrated in tanks experiments, where significant changes*

are only observed when corals are stressed^{13,14}. These results are still valid to confirm the lack of harm caused by the probiotic inoculation on *P. verrucosa in situ*, which is an urgent risk assessment step that can contribute for science-based frameworks for the safe use of probiotics for wildlife¹⁹. More importantly, even if there was nothing to be fixed in the coral health and the microbiome was stable, continuous inoculations provide a microbiome (and, potentially, epigenomic³⁷) restructuring that could be beneficial in times of stress. In addition, the observed variations in Fv/Fm at different sampling times do not appear to be influenced by the inoculation, given that the control samples exhibit a similar trend. Hence, the significant temporal differences seem to be more closely connected to the corals' seasonal responses to temperature changes rather than to the experimental treatment. It is a well-established fact that Fv/Fm ratios fluctuate with the seasons, in correlation with alterations in light and temperature conditions, as documented in the literature^{111,112}. Notably, the Fv/Fm values for both the control and the probiotic-treated corals are indicative of healthy corals. Further *in situ* experiments should investigate the effects of these inoculations in the event of a bleaching event or disease outbreak, and/or the effects of probiotics on diseased or thermally stressed corals displaying signs of bleaching, which would provide insights into the effects of coral probiotics in coral hosts with various health statuses. (Lines 356-381).

PS: It is also important to clarify, as stated in Supplementary Table S1, that the strains used as putative probiotics in this study did not exhibit *in vitro* antagonistic effects against *Vibrio corallilyticus*, nor did they demonstrate urease activity. We hope these additional details provide further clarification into this aspect.

R: Major comment 3

This study conducted CBASS experiments on the same 30 colonies over an 8-month period. This is a great achievement. The study clearly shows that the ED50s change significantly through time, by almost 2C. That is a huge result, as it helps us to understand how flexible the CBASS response is – a potentially major limitation to determining coral thermal tolerance. Again, this should be a main result. The only other explanation is that there are issues with the relevance of the rapid thermal shock response of a CBASS experiment in capturing a relevant measure of thermal tolerance. If that is the case, it should be explored in the discussion.

A: Thank you for your critical examination of our results. We have a number of studies that are not published yet that specifically address the CBASS reproducibility and seasonal variation. In short, CBASS assays (if conducted properly) are highly reproducible (side-by-side CBASS runs of the same colonies produce almost identical rankings and ED50s). As to the change over time: the consistency of ED50s over time seems to be species specific, according to data obtained in other studies from our lab (under review). Some species such as *Porites* sp. look largely identical over time/seasons, while other species such as *Pocillopora* sp. and *Stylophora* sp. seem to acclimate their thermal thresholds over the annual seasonal cycle. We fully agree that these are important considerations that need to be made explicit so they can be considered in research and field applications. We have addressed this as follows (Lines 374-372):

“It is a well-established fact that Fv/Fm ratios fluctuate with the seasons, in correlation with alterations in light and temperature conditions, as documented in the literature^{111,112}. Notably, the Fv/Fm values for both the control and the probiotic-treated corals are indicative of healthy corals. Further *in situ* experiments should investigate the effects of these inoculations in the event of a bleaching event or disease outbreak, and/or the effects of probiotics on diseased or thermally stressed corals displaying signs of bleaching, which would provide insights into the effects of coral probiotics in coral hosts with various health statuses.” (Lines 374-381).

And

“We assessed the *in situ* photosynthetic efficiency of *Symbiodiniaceae* and thermo-tolerance response in CBASS experiments as indicators of coral holobiont health. Interestingly, we observed that the ED50s of the investigated coral species changes significantly through time, by almost 2 °C. Additional surveys expanding the number of sampling points and coral species investigated are needed to elucidate the seasonality of ED50 values. Despite these natural variations, we did not observe any significant changes between treatments in the tested health indicators at any of the sampling times, likely due to the overall healthy status of the colonies throughout the experiment. Healthy microbiomes are more difficult to change⁵² and the probiotic-promoted protection against stress cannot be tested without stress, as it has been demonstrated in tanks experiments, where significant changes are only observed when corals are stressed^{13,14}.” (Lines 356-365).

R: Major comment 4

There is no mention in the methods of when samples were collected (for CBASS, *in situ* Fv/Fm, and microbiome) in relation to when the colonies were inoculated with probiotic treatments. Sampling just after inoculation would of course be a big problem. If that is not the case, there is no problem, it should just be stated. If that is the case, then a clear discussion of this limitation is required. For instance, could the result that there was only an effect at T3 be due to sampling just after the probiotic treatment was applied? This must be clarified

A: Thank you for pointing this out. The sampling of coral fragments (for microbiome analysis and CBASS experiments) was conducted before the colonies were inoculated with probiotics, at the different sampling points (T1-T4). We clarified this in the methods section as follows (Lines: 518-521)

“Fragments from each colony were collected for coral-associated bacterial community analysis by Scuba diving at all sampling times (T1-T4), and before any probiotic inoculations at the moment of sampling, using sterile gloves and pliers (one for each treatment) and individual sterile collection bags (Whirl-Pak®).”

R: L21-23: This is an overstatement. While the authors are talking about only the “treatments”, there are a whole host of other mechanisms that reef managers use to ensure that native corals are not lost. Here, retain, may also be the wrong word to use. Consider alternative to state: “... can enhance the stress resistance of corals in laboratory trials and is now being considered as a treatment for stressed corals in the wild.”

A: We re-phrased the statement taking into account this suggestion, as follows (Lines 20-21):

“Beneficial Microorganisms for Corals (BMCs), or probiotics, can enhance coral resilience against stressors in laboratory trials, and it is now being explored as a treatment strategy in the wild”.

R: L23: should read as “To elucidate this, we inoculated *Pocillopora verrucosa* colonies at a Red Sea reef with BMC treatment three times per week for a three-month period”

A: We re-phrased as follows (Lines: 23-25):

“As a first step to elucidate this, we inoculated putative probiotic bacteria (pBMCs) on healthy colonies of *Pocillopora verrucosa in situ* in the Red Sea, three times per week, during three months.”

R: L24: The intensity of treatment application must be provided in the abstract. Otherwise, it sounds like a one-off treatment may be enough, and risks overselling the result and being blown out of proportion by the media etc. that will only read the abstract.

A: This information was included in the abstract and addressed in the previous comment.

R: L25: “appeared” this choice of wording does not inspire confidence in the result. If there was a robust experimental design, and there was statistically no change, then state that the microbiome of surrounding areas “remained” unchanged.

A: Good point. We changed the word “appeared” with “remained” as follows (Lines 25-26):

“pBMCs significantly influenced the coral microbiome, while those of the surrounding seawater and sediment remained unchanged”

R: L28-29: Makes it sound like this was circumstantial. If it correlated with decreased vibrio, state that more clearly: “... and a decrease in potential coral pathogens, such as *Vibrio*.”

A: Another good point, thank you. We re-phrased the sentence, taking into consideration the suggestion, as follows (Lines 27-29):

“Furthermore, probiotic treatment correlated with an increase in other beneficial groups (e.g., *Ruegeria* and *Limosilactobacillus*), and a decrease in potential coral pathogens, such as *Vibrio*”.

R: L30-32: This statement doesn’t make sense as it should say what exactly the data indicate about feasibility. Notably, given that the microbiome effect was gone after a few months, given that it took 3 months to achieve an effect, and given that there was no impact on thermal tolerance, I think the main message of this abstract should highlight the limitations of probiotics and state the need for more research.

A: We appreciate your comments regarding potential limitations in this study. As previously mentioned, the primary focus of this study was not to induce a physiological effect (or we would not inoculate very healthy corals), but to concentrate on the microbiome changes promoted by probiotic inoculation. We did, however, include the caveats and limitations in the abstract, in lines 29-33:

“As all corals (treated and non-treated) remained healthy throughout the experiment, we could not track health improvements or protection against stress. Our data indicate that healthy, and therefore stable, coral microbiomes can be restructured *in situ*, although repeated and continuous inoculations may be required in these cases.”

R: L39: Neither of these citations are independently peer reviewed. One is an editorially reviewed report and the other is report. Should cite the primary literature for this statement.

A: We changed the references for primary literature relevant for this statement (Constanza et al., 2014; Voolstra et al., 2021; Moberg & Folke, 1999).

R: L41: should say “enhancing the natural resilience of corals to cope”

A: We re-phrased the sentence accordingly, as follows (Lines 41-43):

“there has been a growing focus on active intervention strategies aimed at enhancing the natural resilience of corals to cope with the already established negative effects of environmental stressors^{2,8,9}”

R: L50: rapid adaptation (i.e., increase in frequency of heat tolerance related genes in a population) occurs through genetic means, over multiple generations. Please either state how changing symbiotic partners can help corals adapt, or reword the sentence.

A: We changed the word “adaptation” for “acclimation” (Line 51) as it now provides clarity on the concept avoiding misleading of the key message of the sentence. Bacterial communities can indeed change (referring here as acclimate) in response to changing environmental conditions, as reported in literature (Reshef et al., 2006; Voolstra & Ziegler, 2020).

R: L66: This is where the “three times per week” should be stated. For example: “In this study, we inoculated colonies of the scleractinian *Pocillopora verrucosa* (Ellis and Solander, 1786) with probiotic treatments three times per week over a three-month period to investigate ...”

A: We added the requested information to the sentence, as follows (Lines 66-69):

“In this study, we conducted repeated *in situ* probiotic inoculations three times per week, over a three-month period on colonies of the scleractinian *Pocillopora verrucosa* (Ellis and Solander, 1786) to investigate changes in their microbiome, surrounding microbial communities and coral health.”

R: L69: should read as “an overall restructuring of the coral microbiome”

A: We modified the sentence accordingly, as follows (Lines 69-71):

“Three of the inoculated pBMC genera were enriched in the coral microbiome, which was aligned with an overall restructuring of the coral microbiome.”

R: Figure 1 C – very hard to read dates, consider just showing the 1st of each month in a simple format, rather than the entire dates. Still can keep the minor ticks as weekly.

A: We modified the dates of the figure accordingly for more readability.

R: Figure 2 – Show statistical results here too (only significant difference at T3) and show the percentage variance explained by different MDS axes.

A: We have added the statistical results to the nMDS, showing the significant differences found in T3. As we used nMDS method of ordination, it does not have a percentage of variance in the axes.

R: Figure 3 – were corrections for multiple testing conducted?

A: We conducted single comparisons between treatments for each pBMC, using the Wilcoxon test.

R: L193: should say “yet remained stable over time” as it is in comparison to the previous sentence, given the usage of “similarly” in this sentence.

A: We added your suggestion to the sentence, as follows (Lines: 211-212):

“Similarly, the bacterial community in the sediments was not significantly changed by treatment, yet remained stable over time”

R: L214 missing a close bracket

A: We added the missing bracket (Line 233).

R: L273-274: This study showed completely the opposite – that the new community did not impact thermal stress tolerance at all.

A: Thermal tolerance was not tested, as all corals were healthy and therefore not exposed to stress. We agree however that the sentence was misleading, and re-phrased as follows (Lines 303-311):

“Moreover, we observed a decrease in bacteria from the family *Vibrionaceae*, which encompasses a wide range of marine bacteria, extensively associated to coral microbiomes ⁶⁶. Some *Vibrionaceae* members constitute opportunistic coral pathogens, likely contributing to coral diseases and bleaching ^{43,44,92–99}. We observed that the probiotic treatment led to a decrease in the abundance of *Vibrio* spp., Unclassified *Vibrionaceae*, *Photobacterium*, and *Catenococcus*. Previous studies have shown that probiotics can reduce *Vibrio* abundance and mitigate coral bleaching ^{13,14}. Nonetheless, it is important to note that some *Vibrios* can be non-pathogenic ⁶⁹, and therefore, further studies exploring the restructuring of *Vibrionaceae* with targeted approaches would help enlighten specific shifts in pathogenic and non-pathogenic coral-associated bacteria.”

R: 290-295 – This is an interpretation beyond what the results show. In fact, the results show that inoculating corals with probiotics that improve stress tolerance in the lab do not improve stress tolerance in the field. Therefore, the advice should be the opposite – that rehabilitation projects should be very careful before relying solely on probiotics – as it shows a limited capacity for success in a field trial. This type of over-reaching should not be included

A: We addressed this point in the discussion as it is connected to the major comment # 1, (Lines 356-377).

R: L308: Here the authors should state the message of Major Comment 1 above.

A: We have addressed this major comment through the discussion.

R: L321: The assumption at the beginning of a heat stress assay should be that all corals included are healthy (unless there are hypotheses about prior health on thermal performance – not the case in this study). Therefore, this is not a reason to discount the results of CBASS experiments.

A: As stated above, all corals were healthy during the experiment and therefore, the different sampling times. We wanted to test if the acclimated corals (treated with probiotics) would present any different behavior. When we performed the CBASS experiment the corals responded to stress in the same way (no treatment differences in ED50) in every sampling time. This is similar to the data obtained from tank trials available in the literature, where probiotic improvements are not observed for healthy corals, only for stressed ones.

R: L324: Suggest that probiotics do not seem to cause harm: surely the hope is that probiotics would be beneficial?

A: Good point. Yes, this is the hypothesis, but we are not advocates for a particular approach, only reliable data can ensure that. One of the risk assessment steps to be tested (Peixoto et al., 2022) was off-target or unintended effects. This study addresses this topic, and this needs to be highlighted. We addressed this comment as follows (Lines 365-368).

“These results are still valid to confirm the lack of harm caused by the probiotic inoculation on *P. verrucosa in situ*, which is an urgent risk assessments step that can contribute for science-based frameworks for the safe use of probiotics for wildlife ¹⁹.”

R: L326: The CBASS experiment is a ‘time of stress’, and the results show no effect.

A: Agreed, but at the same time it shows “no stress”. As it is not possible to fix something that is not broken, and the goal of this study was to address risk assessment steps and microbiological questions highlighted in the scientific framework proposed by Peixoto et al., 2022 - Nature Microbiology.

R: L446: should now state when samples were collected in relation to when the probiotic treatment was applied. It is a problem if the probiotic treatment is applied just before the microbiome samples was taken.

A: We added this information to clarify this aspect (Lines 518-521):

“Fragments from each colony were collected for coral-associated bacterial community analysis by Scuba diving at all sampling times (T1-T4), and before any probiotic inoculations at the moment of sampling, using sterile gloves and pliers (one for each treatment) and individual sterile collection bags (Whirl-Pak ®)”.

Reviewers' comments:

Reviewer #1 (Remarks to the Author):

The authors have addressed and answered all the points raised to my complete satisfaction. I have no further comments.

Reviewer #2 (Remarks to the Author):

I have looked over the responses to my previous comments as well as the revised manuscript. I thank the authors for all the hard work they have put into the revisions and I don't have any further comments on the paper.

Reviewer #3 (Remarks to the Author):

Overview

I am sorry for the slow review of the revised version of the paper and appreciate the work you have put into this manuscript. Unfortunately, there are still some major limitations with the study design to answer the questions posed as the primary research questions, and the interpretation of thermal stress assays. Therefore, my recommendation is that the manuscript should not be published in its current form. I hope that my comments can be constructive toward revising the framing of the study. I think this study is a perfect design to test whether the microbiome of corals can be changed in situ (rather than in the lab), and that this should be the main focus of the paper. I still think that more work is needed on the interpretation of the thermal stress assays. However, the main problem I see is that if one were to truly test whether probiotic treatments would have unintended consequences on the microbiome of the environment (in sediments or water), then treatments would need to be conducting on many more than 15 colonies. If this treatment were to be put into practice for conservation or rehabilitation, then presumably hundreds of coral colonies would be targeted. As such, the scale of this test on N=15 treatment colonies is not appropriate to truly determine with scientific assurance whether probiotics treatments would truly have no "permanent off-targeted effects in the surrounding environment". Therefore, I think the title and main aim of the paper to test for off-target effects should be discussion with more caveats regarding treatment intensity in this experimental study (compared to what may occur in a conservation initiative).

Overall Recommendation:

In short, I recommend to (1) focus the paper on the effect of treatment on coral microbiome in situ, (2) tone down the conclusions (and hypotheses) about off-target effects (as to test this would require a different experimental design), and (3) include a clearer discussion of why the treatments did not affect thermal tolerance (as tested for robustly using the thermal tolerance assays).

Major comment 1:

The study seems to be a test for unintended effects of the microbiome detectable in the surrounding environment. However, the sample size of treated corals (N=15) is very low, as such I think there is no surprise that there is no effect on the environment (although good that your team can show this). However, if we were to implement a probiotic treatment in the wild or in a nursery, the target would be much more than 15 corals, perhaps hundreds. Maybe then there would be unintended consequences. Therefore, I am unsure the study design is robust enough to test for unintended environmental consequences, and publishing a paper that over-reaches to the claim that the treatments do not affect the environment is risky. I recommend including more caveats in the result that that there was no effect on the environment, and say that at larger conservation-relevant scales, these treatments could well have an effect and that more research is needed to test this. This fact also would require a change of the title to remove the reference to effects on the surrounding environment,

perhaps just "Probiotics reshapes the coral microbiome in situ"

Major Comment 2 - regarding the lack of an effect of treatments on the ED50 of colonies

The authors have responded with the argument that it is because the corals were healthy that there was no effect. Yet the paper asserts that applying probiotic treatments in advance of a heatwave is a potential strategy. Usually, the cool season comes in advance of heatwaves – as such, corals during this time will also be healthy. Therefore, I think that this argument needs to be considered more carefully and revised. In addition, the result should be added to the abstract to state that "in situ inoculation of BMCs had no effect on coral thermal tolerance thresholds (ED50s)." There is a fundamental problem with the reasoning for the authors point that "Healthy microbiomes are more difficult to change 52 and the probiotic-promoted protection against stress cannot be tested without stress, as it has been demonstrated in tanks experiments, where significant changes are only observed when corals are stressed 13,14". Firstly, the authors have managed to change the microbiome (even if it is healthy), and second the authors have in fact conducted the required stress test experiments (CBASS). The fact that there was no effect of the new microbiome on the thermal stress response needs to be dealt with up front, which the authors have still not done.

The authors state in their rebuttal that "Thermal tolerance was not tested, as all corals were healthy and therefore not exposed to stress.", however, they state in the methods that "Here, CBASS was used as a proxy to assess coral health and determine if the long-term inoculation of coral probiotics had an effect on coral thermal tolerance". These inconsistencies present a very confusing picture of the study design, initial objectives, and results. I think it is fair to say that there is enough data in the paper to comment on whether probiotic treatments were effective at improving the coral thermal tolerance threshold (as stated in their latter statement). For instance, add to the text: "Notably, the Fv/Fm values for both the control and the probiotic-treated corals are indicative of healthy corals." with something like "indicating that the probiotic treatments had no effect on the rapid thermal shock response of corals."

L51 still reports on adaptation (see comment in previous review). R: L50: rapid adaptation (i.e., increase in frequency of heat tolerance related genes in a population) occurs through genetic means, over multiple generations. Please either state how changing symbiotic partners can help corals adapt, or reword the sentence.

Previous major comment 1: logistical challenges

There is still stronger wording required in the manuscript regarding the feasibility of this intervention to rehabilitate / protect corals, and the fact that when inoculation would begin (months before a heatwave) the corals will already likely be healthy – as they will have had the cool season period (typically before onset of heatwaves). Therefore, the challenge of slow alteration of healthy microbiomes will likely remain. The fact that probiotic interventions will be extremely difficult on most reefs due to logistical challenges needs to be highlighted with stronger wording.

Title: **PROBIOTICS RESHAPE THE CORAL MICROBIOME *IN SITU* WITHOUT DETECTABLE OFF-TARGETED EFFECTS IN THE SURROUNDING ENVIRONMENT**

Reviewer #3 (Remarks to the Author):

Overview

I am sorry for the slow review of the revised version of the paper and appreciate the work you have put into this manuscript. Unfortunately, there are still some major limitations with the study design to answer the questions posed as the primary research questions, and the interpretation of thermal stress assays. Therefore, my recommendation is that the manuscript should not be published in its current form. I hope that my comments can be constructive toward revising the framing of the study. I think this study is a perfect design to test whether the microbiome of corals can be changed *in situ* (rather than in the lab), and that this should be the main focus of the paper. I still think that more work is needed on the interpretation of the thermal stress assays. However, the main problem I see is that if one were to truly test whether probiotic treatments would have unintended consequences on the microbiome of the environment (in sediments or water), then treatments would need to be conducting on many more than 15 colonies. If this treatment were to be put into practice for conservation or rehabilitation, then presumably, hundreds of coral colonies would be targeted. As such, the scale of this test on N=15 treatment colonies is not appropriate to truly determine with scientific assurance whether probiotics treatments would truly have no “permanent off-targeted effects in the surrounding environment”. Therefore, I think the title and main aim of the paper to test for off-target effects should be discussion with more caveats regarding treatment intensity in this experimental study (compared to what may occur in a conservation initiative).

Overall Recommendation:

In short, I recommend to (1) focus the paper on the effect of treatment on coral microbiome *in situ*, (2) tone down the conclusions (and hypotheses) about off-target effects (as to test this would require a different experimental design), and (3) include a clearer discussion of why the treatments did not affect thermal tolerance (as tested for robustly using the thermal tolerance assays).

Major comment 1:

The study seems to be a test for unintended effects of the microbiome detectable in the surrounding environment. However, the sample size of treated corals (N=15) is very low, as such I think there is no surprise that there is no effect on the environment (although good that your team can show this). However, if we were to implement a probiotic treatment in the wild or in a nursery, the target would be much more than 15 corals, perhaps hundreds. Maybe then there would be unintended consequences. Therefore, I am unsure the study design is robust enough to test for unintended environmental consequences, and publishing a paper that overreaches to the claim that the treatments do not affect the environment is risky. I recommend including more caveats in the result that that there was no effect on the environment, and say that at larger conservation-relevant scales, these treatments could well have an effect and that more research is needed to test this. This fact also would require a change of the title to remove the reference to effects on the surrounding environment, perhaps just “Probiotics reshapes the coral microbiome *in situ*”

A: We thank you again for your suggestions regarding the environmental effects of coral probiotics applied *in situ*. We agree that the caveats regarding this aspect should go upfront, and have therefore re-framed the conclusion regarding the safety application of coral probiotics *in situ*, highlighting that further studies are required to validate this at more relevant rehabilitation scales, as follows:

Lines 35-37: “Further, our study provides supporting evidence that, at the studied scale, pBMCs have no detectable untargeted effect on surrounding microbiomes of seawater and sediment near inoculated corals, requiring further testing for larger-conservation relevant scales.”

Lines 417-423: “Our results also indicate the lack of detectable changes in the surrounding coral microbiomes, providing supporting evidence of the potentially safe application of coral probiotics in reef ecosystems. The risk assessment and studies on the feasibility and logistics required for larger *in situ* probiotic interventions should be continuously addressed with the upscaling of this rehabilitation strategy. Future studies would also benefit from elucidating the short-term effects of coral probiotics and expanding this research to potential off-target organisms, which would provide an even more comprehensive assessment of the implications and ecological interactions associated with coral probiotic applications in the marine environment at scale. In addition, the protective role promoted by probiotics still needs to be tested in stressed corals *in situ*.”

Major Comment 2 - regarding the lack of an effect of treatments on the ED50 of colonies
The authors have responded with the argument that it is because the corals were healthy that there was no effect. Yet the paper asserts that applying probiotic treatments in advance of a heatwave is a potential strategy. Usually, the cool season comes in advance of heatwaves – as such, corals during this time will also be healthy. Therefore, I think that this argument needs to be considered more carefully and revised. In addition, the result should be added to the abstract to state that “*in situ* inoculation of BMCs had no effect on coral thermal tolerance thresholds (ED50s).” There is a fundamental problem with the reasoning for the authors point that “Healthy microbiomes are more difficult to change 52 and the probiotic-promoted protection against stress cannot be tested without stress, as it has been demonstrated in tanks experiments, where significant changes are only observed when corals are stressed 13,14”. Firstly, the authors have managed to change the microbiome (even if it is healthy), and second the authors have in fact conducted the required stress test experiments (CBASS). The fact that there was no effect of the new microbiome on the thermal stress response needs to be dealt with up front, which the authors have still not done.

The authors state in their rebuttal that “Thermal tolerance was not tested, as all corals were healthy and therefore not exposed to stress.”, however, they state in the methods that “Here, CBASS was used as a proxy to assess coral health and determine if the long-term inoculation of coral probiotics had an effect on coral thermal tolerance”. These inconsistencies present a very confusing picture of the study design, initial objectives, and results. I think it is fair to say that there is enough data in the paper to comment on whether probiotic treatments were effective at improving the coral thermal tolerance threshold (as stated in their latter statement). For instance, add to the text: “Notably, the Fv/Fm values for both the control and the probiotic-

treated corals are indicative of healthy corals.” with something like “indicating that the probiotic treatments had no effect on the rapid thermal shock response of corals.”

A: We see now that our previous response was confusing. When we said we did not test for thermal tolerance, we were referring to the fact that corals were not stressed in their natural environment during the experiment. We apologize for the confusion. All the discussion and conclusions presented for this topic is backed up by our data and also aligned with the available literature.

We have clarified in the text that pBMCs did not have an effect on the rapid thermal shock coral response evaluated in CBASS experiments. We also highlight that the obtained data is aligned with all the literature available on coral probiotics used in tank experiments, where significant differences in the health status of treated and non-treated corals were only observed when the animals were stressed, but not between healthy colonies submitted to different treatments. We have provided the following discussion:

Lines 367-377: “This indicates that the probiotic treatment had no effect on the coral’s rapid thermal shock response, as assessed in CBASS experiments. The rapid response to a thermal shock might be influenced by the overall health status of the host, and the time to acclimate to the stress. Healthy microbiomes are more difficult to change⁵² and even a restructured healthy microbiome might not quickly acclimate to a rapid thermal shock. The potential probiotic-promoted protection against thermal stress might be rather progressive, and evident in corals that are gradually stressed (during days or weeks) as it has been demonstrated in tank experiments that significant differences in the health status of probiotic-treated corals are only observed when corals are stressed, rather than between healthy corals.”

In addition, regarding the point : “Yet the paper asserts that applying probiotic treatments in advance of a heatwave is a potential strategy. Usually, the cool season comes in advance of heatwaves – as such, corals during this time will also be healthy. Therefore, I think that this argument needs to be considered more carefully and revised”, we changed this in the previous version, mentioning the argument that probiotics are likely more effective when applied in stressed corals, therefore, not mentioning applying them in advance before a heatwave, as follows:

Lines 332-335: On the contrary, the enrichment of probiotics seems to be facilitated when corals are under stress¹⁴, which may indicate that, if the goal is the rehabilitation and retention of threatened corals, the use of probiotics as medicine applied in times of stress may be a good strategy to be tested^{10,102}.”

L51 still reports on adaptation (see comment in previous review). R: L50: rapid adaptation (i.e., increase in frequency of heat tolerance related genes in a population) occurs through genetic means, over multiple generations. Please either state how changing symbiotic partners can help corals adapt, or reword the sentence.

A: Thank you for pointing this out. We used the word “acclimation” instead, as it refers to the rapid microbiome changes (that may occur in days or weeks) that a coral holobiont may undergo in response to changing environmental conditions. This includes microbiome flexibility, which might help the coral respond to environmental stress (For example, see Episten et al.,

2019: Microbiome Engineering: Enhancing climate resilience in corals,
<https://doi.org/10.1002/fee.2001>.

***It's noteworthy that the ED50 values for this particular coral species showed seasonal variation (we further explore this question in a different paper Garcia et al, under review). However, we observed no discernible differences between treatments during the experiment. This lack of disparity isn't surprising, given that the corals remained healthy throughout the experiment, as indicated by Fv/Fm data. We anticipate that probiotics could enhance thermal tolerance and/or fitness (Fv/Fm) compared to sick or unhealthy corals, based on our observations during the recent bleaching event in the central Red Sea (Garcias-Bonet at al. in prep). Nevertheless, utilizing probiotics without modifying their thermal traits (e.g., by introducing symbionts adapted to extreme temperatures) shouldn't impact the thermal threshold of healthy corals, as this would require adaptation mechanisms to come into play.

Previous major comment 1: logistical challenges

There is still stronger wording required in the manuscript regarding the feasibility of this intervention to rehabilitate / protect corals, and the fact that when inoculation would begin (months before a heatwave) the corals will already likely be healthy – as they will have had the cool season period (typically before onset of heatwaves). Therefore, the challenge of slow alteration of healthy microbiomes will likely remain. The fact that probiotic interventions will be extremely difficult on most reefs due to logistical challenges needs to be highlighted with stronger wording.

A: Thank you for your comment. Our recent and unpublished data actually indicate that the inoculations may even as well start during the heatwave and still provide significant protection to corals. Unfortunately, we cannot present this data here, and some of the analyses are still being performed before we can submit this other paper. We have, however, added the following sentence for clarification, highlighting the need for further studies addressing those limitations:

Lines 345-349: “The necessary frequency of inoculations (days, weeks, or months) and probiotic cell concentration may depend on different variables, such as the goals of the intervention and environmental conditions, and need further investigation and optimization, aiming at reducing logistics efforts when applying coral probiotics at larger rehabilitation scales.”